# Tunable backbone-degradable robust tissue adhesives via in situ radical ring-opening polymerization

Ran Yang [1,2], Xu Zhang [1], Binggang Chen [1] ✉, Qiuyan Yan[1] ✉, Jinghua Yin[1] & Shifang Luan [1,2] ✉

Adhesives with both robust adhesion and tunable degradability are clinically and ecologically vital, but their fabrication remains a formidable challenge. Here we propose an in situ radical ring-opening polymerization (rROP) strategy to design a backbone-degradable robust adhesive (BDRA) in physiological environment. The hydrophobic cyclic ketene acetal and hydrophilic acrylate monomer mixture of the BDRA precursor allows it to effectively wet and penetrate substrates, subsequently forming a deep covalently interpenetrating network with a degradable backbone via redox-initiated in situ rROP. The resulting BDRAs show good adhesion strength on diverse materials and tissues (e.g., wet bone >16 MPa, and porcine skin >150 kPa), higher than that of commercial cyanoacrylate superglue (~4 MPa and 56 kPa). Moreover, the BDRAs have enhanced tunable degradability, mechanical modulus (100 kPa-10 GPa) and setting time (seconds-hours), and have good biocompatibility in vitro and in vivo. This family of BDRAs expands the scope of medical adhesive applications and offers an easy and environmentally friendly approach for engineering.

Robust adhesives are widely used in the biomedical and the bioelectronic industry[1,2]. In these contexts, achieving good interfacial bonding to wet surfaces has proven to be challenging, especially for complex biological tissue-involved wound management and implantable device/sensor anchors[3]. Recent years have seen transformative advances in achieving good adhesion to wet materials, most widely in biological tissues, through supermolecule and electrostatic interactions[4–6], covalent bonding[7,8], and topological adhesion[9–11]. However, previous wet adhesives mainly show high adhesion energy (up to 1000 J m⁻²) rather than adhesive strength, which needs to be increased from hundreds of kPa to MPa for many applications, such as bone adhesion[12–16]. The adhesion strength of these adhesives, such as hydrogels, has long been limited by the low permeability of polymers (which are sometimes impossible to diffuse and entangle with tissues) and low bulk modulus (mechanical mismatch with high-modulus

substrates)[17–19]. Adhesion is also plagued by the mechanical deterioration of swelled hydrophilic polymer networks in the wet environment[20–22] (Fig. 1a). Unlike polymer adhesives, small molecule adhesives with better permeability generally have a much higher adhesion strength[23]. Cyanoacrylate (CA) superglue has strong adhesion due to the good diffusion of the monomer solution during rapid in situ polymerization (Fig. 1b). Whereas, this glue prematurely forms a stiff and brittle adhesive layer due to the instant anionic polymerization initiated by water, resulting in weak bonding between wet and soft tissues. Furthermore, CA is hardly degradable and toxic[24]. Significantly, non-degradability greatly limits the medical applications of adhesives due to additional surgical removal requirements and tissue regeneration obstacles and causes enormous environmental pressure[25,26]. However, most degradable adhesives fabricated with biomolecules or biobased polymers (e.g., fibrin, gelatine, or alginate) only have limited

[1]State Key Laboratory of Polymer Physics and Chemistry, Changchun Institute of Applied Chemistry, Chinese Academy of Sciences, Changchun 130022, China. [2]School of Applied Chemistry and Engineering, University of Science and Technology of China, Hefei 230026, China. ✉e-mail: bgchen@ciac.ac.cn; qyyan@ciac.ac.cn; sfluan@ciac.ac.cn

cohesion strength because they have poor structural homogeneity and poor mechanical properties[27–30]. Thus, robust adhesives with tunable degradability remain a formidable challenge due to the intrinsic trade-off between adhesive stability and strength.

Achieving high adhesive strength and degradability requires combining two aspects: (I) Well-permeated precursors cured to form a cross-linked network with good mechanical properties to obtain robust cohesion and interfacial bonding. (II) The backbone of the in situ formed polymeric network has biodegradable units. Marine organism mussels have good adhesion to wet substrates due to the secretion of mussel foot proteins consisting of hydrophilic and hydrophobic polypeptide chains[31]. The hydrophilic parts promote interfacial penetration, and the hydrophobic parts facilitate stable adhesion[32]. Radical ring-opening polymerization (rROP) is a kind of ring-opening polymerization initiated by radicals as active species[33]. rROP efficiently combines the simplicity and robustness of the radical polymerization of vinyl monomers with the controlled degradable chain of ring-opening polymerization of lactones, cyclic carbonates, or N-carboxyanhydrides, so it has been utilized to pre-synthesize biodegradable polymers for further biomedical applications, e.g., biodegradable micelles/nanoparticles and functional polyester sealing materials[34,35].

Herein, inspired by the synergistic roles of the hydrophilic and hydrophobic chains of mussel foot proteins, an in situ rROP of hydrophobic cyclic ketene acetal (CKA) monomers and hydrophilic acrylate comonomers in the physiological environment was proposed to tunable synthesize a backbone-degradable robust adhesive (BDRA) (Fig. 1c). The amphipathic BDRA precursors are well wetted before curing and achieve high diffusion-dominated interpenetration toward adherents with different surface energies, even for dense biological tissues. Noteworthy, the rROP is initiated by a redox system to in situ form backbone-degradable polymer adhesive without being affected by environmental factors, including water (Fig. 1d). Ultrastrong adhesion is achieved upon the backbone-degradable covalent interpenetrating network solidified in a wide setting window ranging from seconds to hours (Fig. 1e). Benefiting from the flexibility and controllability of the rROP, the degradability and mechanical properties of the BDRAs can be customized on-demand, and their overall performance compares favorably to that of sixteen well-recognized adhesives[36–39] (Fig. 1f and Supplementary Table 1), providing a broad spectrum of possibilities for biomedical engineering and medical applications in a facile and environmentally friendly manner.

## Results and discussion

### BDRA designs and adhesive performances

Taking advantage of the rROP strategy, a family of BDRAs was facilely fabricated by copolymerizing 2-methylidene-1,3-dioxepane (MDO, a typical CKA monomer with complete ring-opening tendency[40]) with eighteen acrylate comonomers at room temperature without additional water and oxygen removal (Supplementary Table 2). To evaluate the adhesion performance of the BDRAs fabricated by in situ rROP, hydroxyethyl methacrylate (HEMA) and hydroxyethyl acrylate (HEA) were selected as the hydrophilic acrylate comonomers of the BDRAs (Supplementary Fig. 1), and three types of substrates (i.e., biological tissues, polymers, and metals) were separately adhered to by the BDRAs and commercial adhesives.

The adhesion strength of BDRAs on wet bone was >16 MPa by the standard flexural test (~4 MPa for CA references, ~0.2 MPa for Coseal, ~0.1 MPa for Fibingluraas), and that on porcine skin reached ~ 150 kPa by the standard shear test (~60 kPa for CA references, ~20 kPa for Coseal, ~10 kPa for Fibingluraas) (Fig. 1h). The BDRAs exhibited stronger adhesion to biological tissues than commercial tissue adhesives, including the CA adhesives Vetbond, Compont, Baiyun and Dermabond, the polyethylene glycol adhesive Coseal, and the fibrin adhesive Fibingluraas (Supplementary Fig. 2). To visualize the robust

adhesion, we lifted a 60 kg weight with a fractured bovine bone bonded by the BDRAs (MDO-HEMA family) (Fig. 1g and Supplementary Movie 1). A pigskin with an adhered area of $10 \times 15\,mm^2$ bonded by the BDRAs (MDO-HEA family) was used to lift a bucket of 2.7 kg water (Supplementary Fig. 3). The thermal effect of BDRAs was evaluated during in situ curing on pig skin (Supplementary Fig. 4). The tested temperatures lower than 45 °C were observed for the BDRA (MDO-HEMA) family, which were lower than that of Vetbond (~54 °C) and the thresholds for bone necrosis (56 °C)[41]. The tested temperatures of the BDRA (MDO-HEA) family ranging from 39 °C to 59 °C, were much lower than that of the acrylate homopolymer system (~100 °C), suggesting a lower damage probability of skin[42].

Additionally, the adhesion strength of the BDRAs on low surface-energy polymers, including PP, PE, and PTFE, reached 421, 265, and 114 kPa, respectively. They were higher than those of commercial engineering adhesives, such as the CA-based adhesive PR100 (~5 kPa for PP, ~63 kPa for PE, ~2 kPa for PTFE) and the polyurethane-based adhesive 6310 NS (~25 kPa for PP, ~53 kPa for PE, ~48 kPa for PTFE) (Fig. 1i). For the other nine common polymeric and metal substrates, the adhesion of the BDRAs was also comparable to that of these commercial adhesives (Supplementary Fig. 5).

### Adhesion mechanism of BDRAs

Considering the innovative application of in situ rROP with amphipathic small molecule precursors, we speculated that the above strong adhesion of BDRAs to various biological tissues and engineering substrates stemmed from the good wettability and high penetration of the amphipathic BDRA precursors, followed by the formation of a deep covalent interpenetrating network with substrates through the in situ rROP (Fig. 2a). Therefore, BDRAs essentially break through the limitations of CA superglue in terms of wet adhesion (low penetration due to water-initiated rapid anionic polymerization) (Fig. 2b).

The wettability of BDRAs to diverse materials was first studied. Supplementary Fig. 6 shows that BDRA precursors (hydrophobic MDO and hydrophilic acrylate comonomer mixtures without initiator) exhibit tunable surface tension and better wettability to twelve surfaces compared to a water drop. In particular, the BDRA precursors could wet low-energy surfaces, achieving a much greater degree of molecular-level interfacial interaction, unlike CA with poor wettability to these materials, thus leading to weak bonding[24] (Fig. 2c). By mixing isothiocyanate fluorescein, the tissue permeability of BDRA precursors without initiator and BDRA precursors with initiator accompanied by chain propagation during the in situ rROP were comparatively investigated. The penetration depth of the BDRA precursors (MDO-HEA) without initiator into porcine skin increased with time and reached ~200 μm in 60 min, >6 times that of CA (Fig. 2d and Supplementary Fig. 7). A comparison of permeability between different hydrophilic and hydrophobic monomers demonstrated that the amphiphilic BDRAs consisting of hydrophobic MDO segments and hydrophilic HEA segments had higher adhesion strength and thicker bonding interfaces on porcine skin adhesion than hydrophobic poly(MDO-co-(butyl acrylate)) (P(MDO-BA)) and hydrophilic PHEA (Fig. 2e and Supplementary Fig. 8). Similar results were obtained for bone tissue adhesion, in which amphiphilic poly(MDO-HEMA) achieved >6 times higher adhesion strength than hydrophilic PHEMA (Fig. 2f). This was probably due to the similarity between the amphiphilic properties of BDRAs and the characteristics of biological tissues. Based on the good wettability and deep penetration demonstrated above, the adhesion strength of BDRAs on bovine bone increased with time, mainly owing to the formation of a continuously thickened and strengthened bonding interface, and it increased to ~2 MPa within 10 min (Fig. 2g). The above positive correlation between adhesion strength, penetration depth, and penetration time, along with the compact bonding interface, confirmed the formation of a topologically entangled interpenetration polymer network for robust adhesion[9,43].

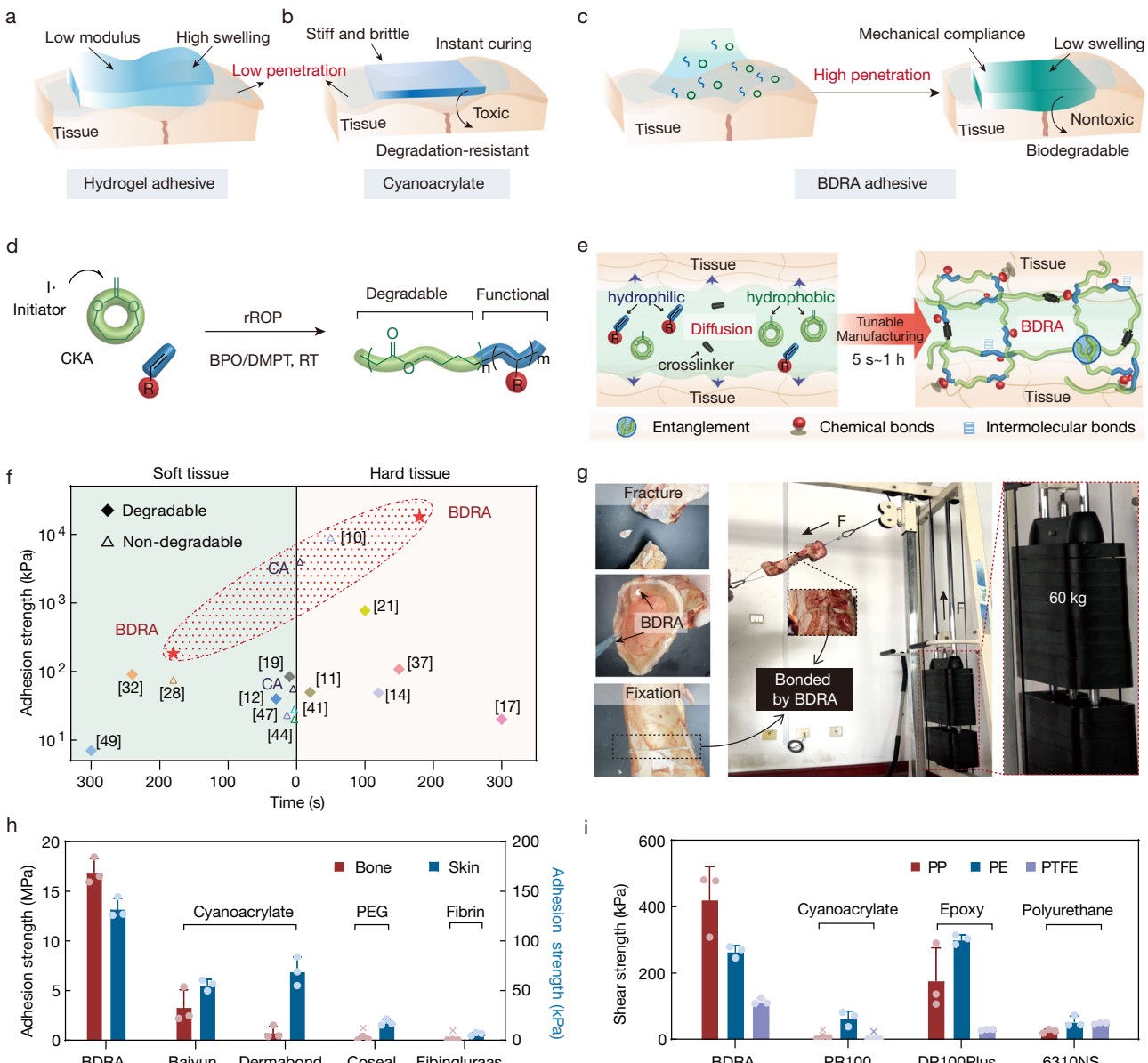

**Fig. 1 | Design and adhesion performance of BDRAs. a–c** Schematic illustrations of the adhesion to wet tissues of hydrogel adhesives (**a**), CAs (**b**), and BDRAs (**c**). **d** In situ rROP of CKA and comonomers to form a degradable and functional macromolecular chain by redox initiation benzoyl peroxide/N,N-dimethyl-p-toluidine (BPO/DMPT). **e** Tunable preparation of BDRAs that achieve strong adhesion by forming a covalent interpenetrating network by in situ rROP and the synergy of intermolecular and chemical bonds. **f** Adhesion strength and setting time of BDRAs and the existing tissue adhesives for hard and soft tissues. [*n*] is the reference number. **g** Bearing capacity of bonded fractured bovine bone using BDRAs. **h** Adhesion strength of BDRAs and commercial medical adhesives on different biological tissues, represented by flexural strength for bone and shear strength for pigskin. Data are presented as the means ± SDs, *n* = 3 independent samples per group. **i** Shear adhesion strength for low-surface-energy polymers adhered by a BDRA and commercial engineering adhesives. PP polypropylene, PE polyethylene, PTFE polytetrafluoroethylene. Data are presented as the means ± SDs, *n* = 3 independent samples.

The swelling ratio of amphiphilic BDRAs was approximately 3.8% in PBS, nearly 10 times lower than that of PHEMA (Fig. 2h), and the BDRA-bonded stainless sheets remained stable for 48 h underwater without swelling-induced deformation (Fig. 2i). Interface failure occurred for the hydrophilic PHEMA within 8 h due to high swelling and was accompanied by a decreasing bonding strength (Fig. 2j and Supplementary Fig. 9). Moreover, reactive groups could be facilely introduced into the BDRAs in the form of functional acrylate comonomers as needed, allowing the synergistic adhesion of chemical bonding and topological entanglement. According to previous reports[7,44,45], acrylic acid N-hydroxysuccinimide ester (AAc-NHS) was copolymerized to produce BDRAs with NHS ester hanging groups, which were able to covalently bond with the primary amine groups on various tissues within a few minutes. There was no obvious reduction in the adhesion strength of BDRAs with AAc-NHS in 48 h (Supplementary Fig. 10). Namely, both swelling resistance and covalent adhesion also endow BDRAs with more stable adhesion.

## Tunable degradability, mechanical properties, and curing time

In addition to improving the hydrophobicity of the polymer network, the CKA monomer of rROP enables the intermittent introduction of cleavable bonds within the backbone[46]. The degradation of BDRAs was believed to be similar to the generally recognized degradation of CKA-based copolymers that undergo a hydrolytic or an enzymatic degradation process[33,47] (Fig. 3a).

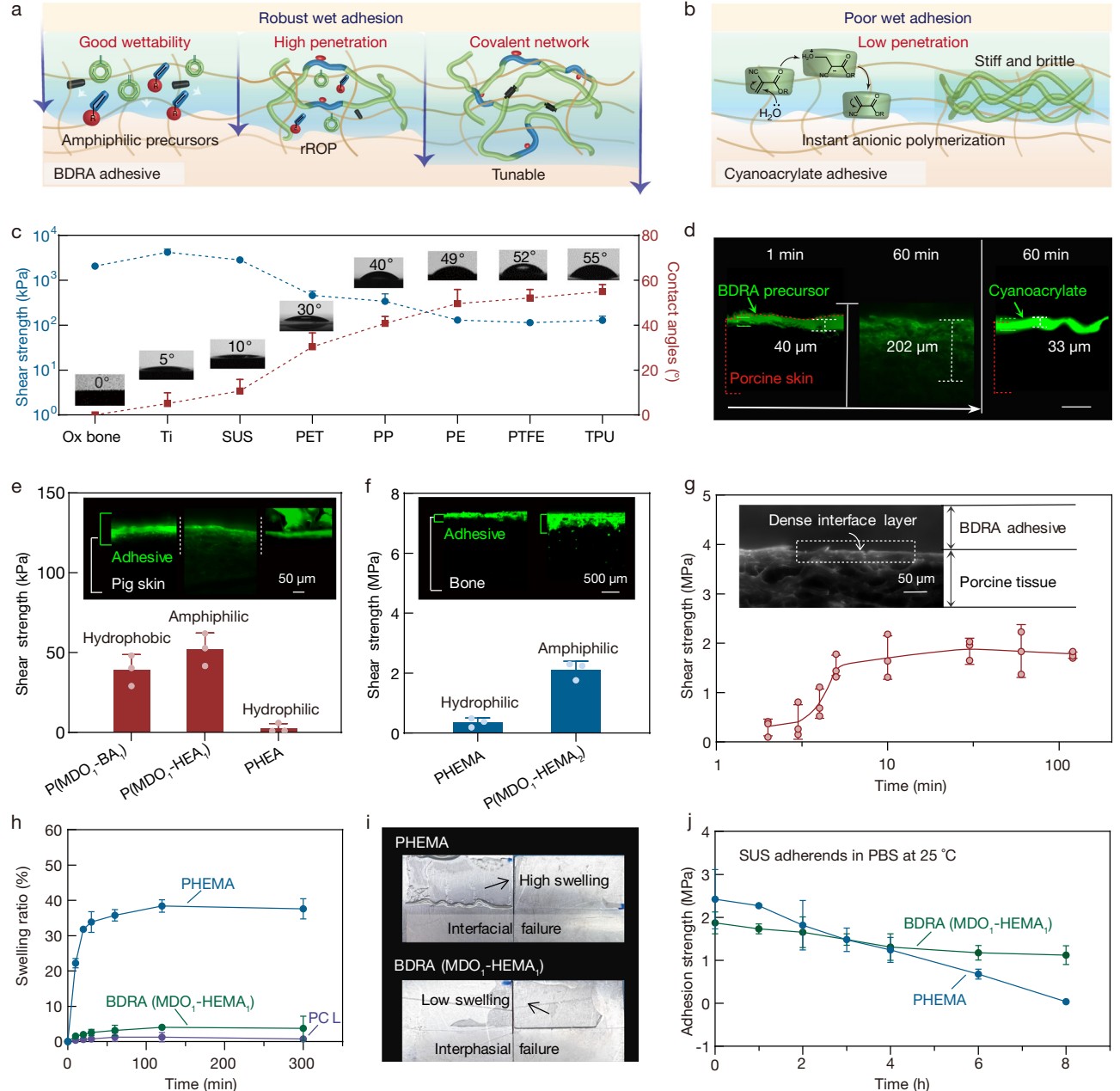

**Fig. 2 | Adhesion mechanism of BDRAs. a**, **b** Schematics of the penetration and curing process of BDRAs (**a**) and CAs (**b**) for wet surface adhesion. **c** Contact angles of the BDRA precursor without initiator on various surfaces versus adhesion strength of the corresponding BDRA. Data are presented as the means ± SDs, $n = 6$ independent samples for contact angles; $n = 3$ independent samples for shear strength. Ti titanium, SUS steel use stainless, PET polyethylene terephthalate, TPU thermoplastic polyurethanes. **d** Fluorescence images of the BDRA precursor without initiator and CA (Vetbond) on porcine skin with time, scale bar = 100 μm. Three independent experiments were conducted with similar results. **e**, **f** Representative fluorescence images and the adhesion strength of amphiphilic BDRAs and their references on porcine skins (**e**) and bovine bones (**f**) under wet conditions. Data are presented as the means ± SDs, $n = 3$ independent samples. **g** Representative dark-field confocal images of the adhesive interface between a BDRA (MDO$_1$-HEA$_1$) and porcine skins and the shear adhesion strength of a BDRA (MDO$_1$-HEMA$_1$) on bovine bone with time. Data are presented as the means ± SDs, $n = 3$ independent samples. **h** Swelling behavior of poly(ε-caprolactone) (PCL) homopolymer, PHEMA, and BDRAs (MDO$_1$-HEMA$_1$) in phosphate-buffered saline (PBS) at 37 °C. Data are presented as the means ± SDs, $n = 3$ independent samples. **i**, **j** Digital photographs (**i**) and adhesion strength with time (**j**) of BDRA (MDO$_1$-HEMA$_1$) and PHEMA on SUS in PBS at 25 °C. Data are presented as the means ± SDs, $n = 3$ independent samples.

The in vitro and in vivo degradation experiments proved that BDRAs consisting of biodegradable polyester and HEMA segments presented much better degradability (Fig. 3b, c). The degradation rates of the composition-optimized BDRAs (MDO$_1$-HEMA$_1$) were 43% after 16 weeks in PBS and 36% after 8 weeks of implantation, which were higher than those of CA (8% and 9%), PCL (1% and 2%) and PHEMA homopolymer (18% and 3%). Interestingly, we found an equilibrium between the contents of MDO and HEMA in the BDRAs. BDRAs (synthesized with an initial MDO feed composition of 0.5) lost 24% of their initial weight after 24 h at 37 °C in 1 M NaOH, which is a greater loss than those of other BDRAs with more and less ester groups (initial MDO feed compositions of 0.75 and 0.25 respectively) (Fig. 3d). We supposed that in addition to the ester bond content, the degradability of BDRAs also benefits from hydrophilicity.

For further confirmation, three types of acrylate monomers with different hydrophilicity were copolymerized with MDO at the same

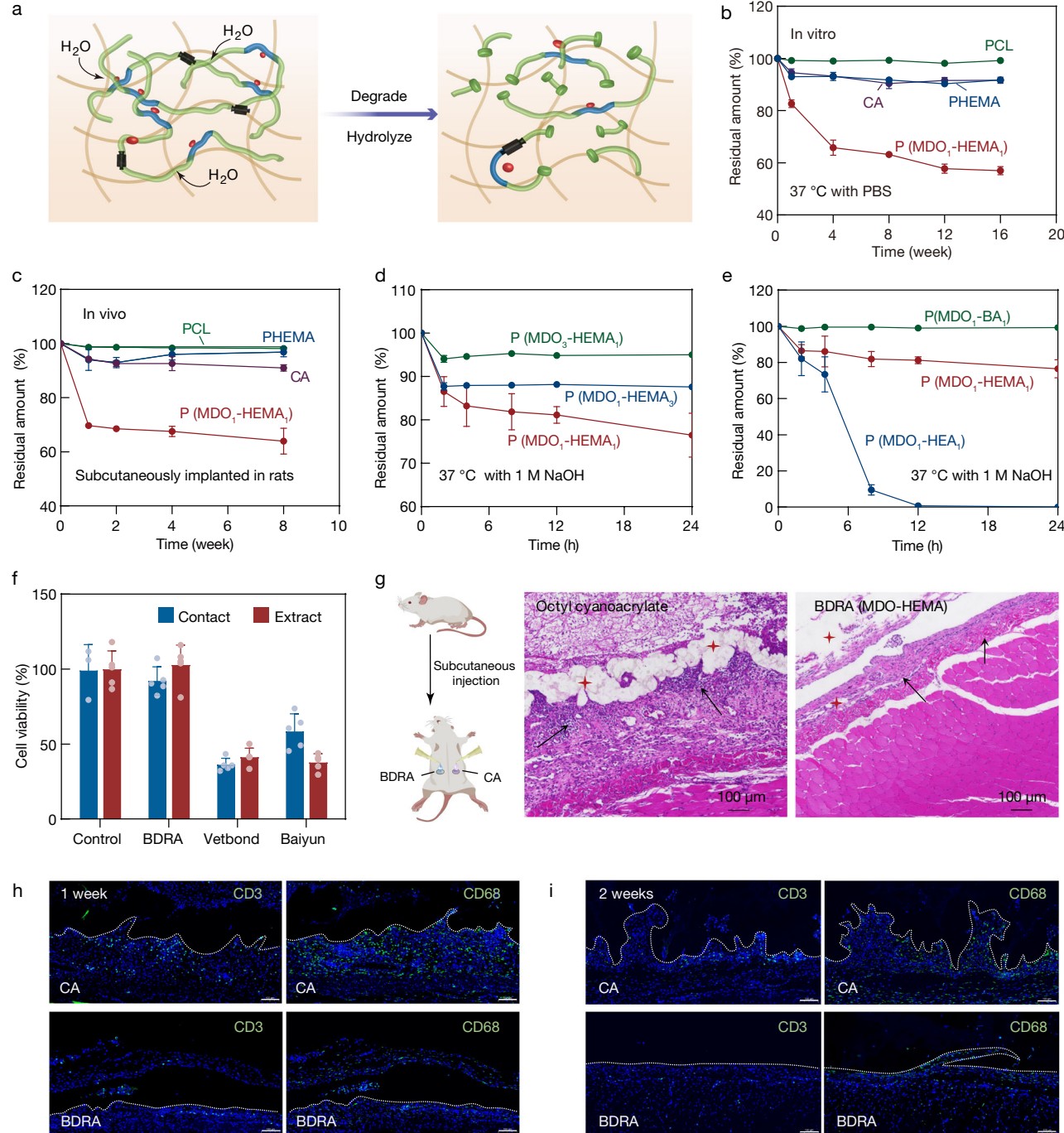

**Fig. 3 | Degradation profiles and biocompatibility of BDRAs. a** Degradation mechanism of CKA-based copolymers. **b** In vitro degradation rates of PCL, PHEMA, and BDRAs (MDO$_1$-HEMA$_1$) based on the remaining weight percentage in PBS at 37 °C. Data are presented as the means ± SDs, $n = 3$ independent samples. **c** In vivo biodegradation behavior of BDRAs (MDO$_1$-HEMA$_1$) implanted subcutaneously in rats. Data are presented as the means ± SDs, $n = 3$ independent samples. **d**, **e** Accelerated degradation profiles of the BDRAs tested in 1 M NaOH solution at 37 °C. Degradation rates vary with the comonomer ratio (**d**) and the comonomer types (**e**) of BDRAs. Data are presented as the means ± SDs, $n = 3$ independent

samples. **f** Cell viability measured by the CCK-8 assay. 2% DEME was used as a control, and the data are presented as the means ± SDs, $n = 5$ independent samples. **g** Representative histological images of BDRAs stained with hematoxylin and eosin (H&E) after abdominal subcutaneous injection in vivo. The red stars represent glues, and the black arrows indicate the presence of inflammatory cells. Immuno-fluorescence staining of lymphocyte (CD3) and macrophages (CD68) after 1 week (**h**) or 2 weeks (**i**) of BDRAs injection. Scale bars, 100 μm. The experiments in (**g–i**) were repeated independently three times with similar results.

initial feeding ratio of 0.5 to fabricate BDRAs. The P(MDO-BA) with hydrophobic BA was degraded by 0.75% after 24 h at 37 °C in 1 M NaOH, a much lower percentage than that of the P(MDO-HEMA) and P(MDO-HEA) with hydrophilic HEMA and HEA, which degraded by 24% and 100%, respectively (Fig. 3e). All of these results indicated that the degradation rates of BDRAs can be adjusted by the dosage and type of

acrylate comonomers and that it is possible to achieve a compatible rate with tissue repair.

Additionally, the implanted BDRAs showed lower cytotoxicity (with cell viabilities >87%) than Vetbond (<60 %) towards osteoblast precursor MC-3T3 cells and mouse fibroblast L929 cells (Fig. 3f and Supplementary Fig. 11). Furthermore, the potential toxicity of residual

monomers of BDRAs was evaluated (Supplementary Fig. 12). The concentrations of residual precursors including MDO, HEMA, NHS, BPO and DMPT presented no potential cytotoxicity according to International Standard ISO 10993-5. Although the highest concentrations of residual HEA were found cytotoxic in the cell tests, preclinical validation in rat abdominal and dorsal implant models demonstrated that the BDRA adhesives, along with their corresponding residual precursors, caused only a milder inflammatory response than octyl CA at an early stage, and the inflammation gradually disappeared (Fig. 3g). As expected, the BDRAs was verified a less proliferating lymphocyte (CD3) and macrophage (CD68) than CA after 7 days (Fig. 3h). And the signs of inflammation for BDRA almost disappeared at 2 weeks, while obviously visible for CA (Fig. 3i). The BDRA adhesives did not cause any noticeable indications of damage or pathological changes of major organs, including the heart, liver, spleen, lung, and kidney, indicating tolerable toxicity and biocompatibility (Supplementary Fig. 13). In addition, Supplementary Fig. 14 shows that BDRAs with the possibility of releasing volatile organic compounds (VOCs) did not affect the normal life activities of mice, and no abnormalities were observed in lung tissue compared to healthy controls.

BDRAs can be regulated to be mechanically compliant with different tissues to ensure strong and stable adhesion. As a representative demonstration, BDRA (MDO-HEA) and BDRA (MDO-HEMA) were designed for soft and hard tissue adhesion, respectively. The glass transition temperature (Tg) of the BDRA (MDO-HEA) adhesives was lower than 37 °C (12–33 °C), and they were in a rubbery state at physiological temperature (Fig. 4a and Supplementary Fig. 15). Moreover, their elastic modulus ranged from $10^2$ to $10^3$ kPa and was similar to that of soft tissue (elastic modulus ~$10^2$ kPa) (Fig. 4b and Supplementary Fig. 16). The Tg of the BDRA (MDO-HEMA) adhesives was larger than 37 °C (50–100 °C), and they were in a glassy state at physiological temperature (Fig. 4c and Supplementary Fig. 17), and their elastic modulus ranged from $10^4$ to $10^6$ kPa, which was similar to that of hard tissues (elastic modulus ~$10^5$ kPa) (Fig. 4d and Supplementary Fig. 18). Because of their good capability of mechanical compliance matching with different biological tissues, the BDRAs exhibited good adhesion performance in various application scenarios.

For pigskin adhesion, the composition-optimized BDRA (MDO-HEA) provided the highest adhesion strength up to 130 kPa by the shear test, 153 kPa by the tensile test, and 131 kPa by the wound closure test (56, 17, and 27 kPa for CA reference); moreover, the interfacial toughness reached 183 J m$^{-2}$ (20 J m$^{-2}$ for CA) (Supplementary Figs. 19 and 20, and Supplementary Movie 2). Additionally, the as-prepared BDRAs were applied to other biological soft tissues and organs, including muscle, heart, liver, spleen, lung, kidney, intestine, and stomach (Supplementary Fig. 21). The BDRAs formed a tight connection with soft tissue benefiting from mechanical matching, in contrast with the obvious gap between the stiff CA and soft tissue, thus effectively avoiding the mechanical stress and inflammatory responses caused by adhesive-to-host incompliance[2] (Supplementary Fig. 22). For wet bovine bone, the composition-optimized BDRA (MDO-HEMA) showed stronger adhesion performance, i.e., in terms of flexural stress (16.97 MPa versus 1.56 MPa), tensile strength (6.46 MPa versus 0.93 MPa) and shear strength (3.05 MPa versus 1.68 MPa) (Supplementary Figs. 23 and 24) than commercial butyl CA (Vetbond).

The adhesion time of BDRAs could also be coordinated according to various medical applications, from instant hemostasis to fracture fixation (setting time windows ranging from seconds to hours)[48,49]. The copolymerization process of MDO with eighteen acrylate comonomers via in situ rROP can reach completion in only a few seconds (e.g., 5 s for HEA, 9 s for AA, 12 s for HPA, comparable to that of the well-recognized rapid curing CA adhesives), dozens of seconds (e.g., 32 s for MA, 45 s for EA, 55 s for PEGMA), several minutes (e.g., 60 s for BA, 2 min for 2-EHA, 7 min for IEM), dozens of minutes (e.g., 12 min for BzMA, 13 min for MPTMS) and a much longer time (Fig. 4e and Supplementary

Table 2). In addition, the curing time could also be flexibly changed by changing the feeding ratio of MDO and acrylate comonomers (Fig. 4f).

## Biomedical applications of BDRAs

We investigated a range of potential applications of BDRAs by a series of ex vivo and in vivo animal experiments. For soft tissue-related clinical applications, BDRAs are particularly advantageous in complex wound treatment because of their convenient implementation. The increased complexity of wounds had little effect on BDRAs (~13 s for line/cruciate incision) and the following wound healing but caused the processing time of suturing and stapling to greatly increase (Fig. 5a). Compared with traditional suturing and stapling, BDRAs facilitated much quicker and tighter closure of skin wounds without mechanical damage (Supplementary Movie 3). Additionally, BDRA showed faster wound healing and less inflammation compared to sutures and CA 7 days after treatment (Supplementary Fig. 25). Besides, BDRAs can effectively seal bleeding by providing fast and strong bio-adhesion. The blood loss of liver perforations in rats sealed by a BDRA was <52 mg, which was less than that of CA (~90 mg) and the commercial hemostatic material Surgicel (~112 mg) (Fig. 5b). Additionally, BDRAs are easy to apply with higher adhesion strength and better stability (compared to Fibingluraas and CA references) and better sealing (compared to Surgicel references) (Supplementary Movie 4). Notably, biodegradable BDRA can be retained after hemostasis to avoid secondary tissue damage and has shown a weak hindrance to the healing of liver tissue damage compared to non-biodegradable CA adhesives (Supplementary Fig. 26). Also, the blood loss of caudal arteries sealed by BDRA was nearly four times lower than that of Fibingluraas, and the rapid and tight sealing by a BDRA for carotid hemostasis effectively prevented death due to untimely sealing (Fig. 5c).

In hard bone-related applications, BDRAs (MDO-HEMA) could conform to irregular defects and effectively fix small fragments and segmental fractures of cattle femur ex vivo (Supplementary Fig. 27). In the physiological environment, the BDRAs bonded comminuted radius fractures in a convenient operation (Fig. 5d) and accurately repositioned the skull fragments of rats within 3 min (Fig. 5e and Supplementary Movie 5). Micro-CT imaging showed significant bone tissue bridging for the BDRA group after 8 weeks, while bone nonunion persisted for the nondegradable CA group and the blank group (Fig. 5f). The degradable BDRA in fractures diminished during bone regeneration and ingrowth and finally almost disappeared over 8 weeks, while the CA adhesive was still significantly observed (Fig. 5g). Quantitative analysis of bone regeneration revealed that the bone volume fraction (BV/BT) of biodegradable BDRA was the highest among all groups, with values of 31.4 %, 50.6%, and 59.6% after 2, 4, and 8 weeks, respectively. In contrast, the BV/BT of nondegradable CA (33.1%) was even lower than that of the blank references (~39.7%) after 8 weeks post-surgery, representing an obstacle to tissue healing (Fig. 5h). Biodegradable BDRA provides a spatial environment for fibrous and osseous tissue ingrowth, allowing for more newly formed bone islands and dense fibrous tissue. (Fig. 5i, j) These results further demonstrated the degradability of BDRAs is essential for tissue healing.

We further showed that BDRA facilitates the integration of biomedical materials/devices onto the surfaces of various biological tissues. BDRAs were used to facilitate the flexible adhesion of polypropylene mesh to abdominal muscles for abdominal hernia treatment, and this mesh did not come off even under twisting motions (Supplementary Fig. 28). BDRAs were also used adhered a pressure sensor to a porcine heart, and allowed fast and flexible attachment and measurement of the dynamic behaviors (Supplementary Fig. 29). Furthermore, BDRAs enabled the firm fixation of titanium alloy stents and implants, polylactic acid bone nails, and polyether ether ketone bone nails to bone and muscle tissues, regardless of the presence of blood (Supplementary Fig. 30).

In this study, we proposed a design strategy for tunable backbone-degradable robust adhesives by in situ rROP of hydrophobic CKA

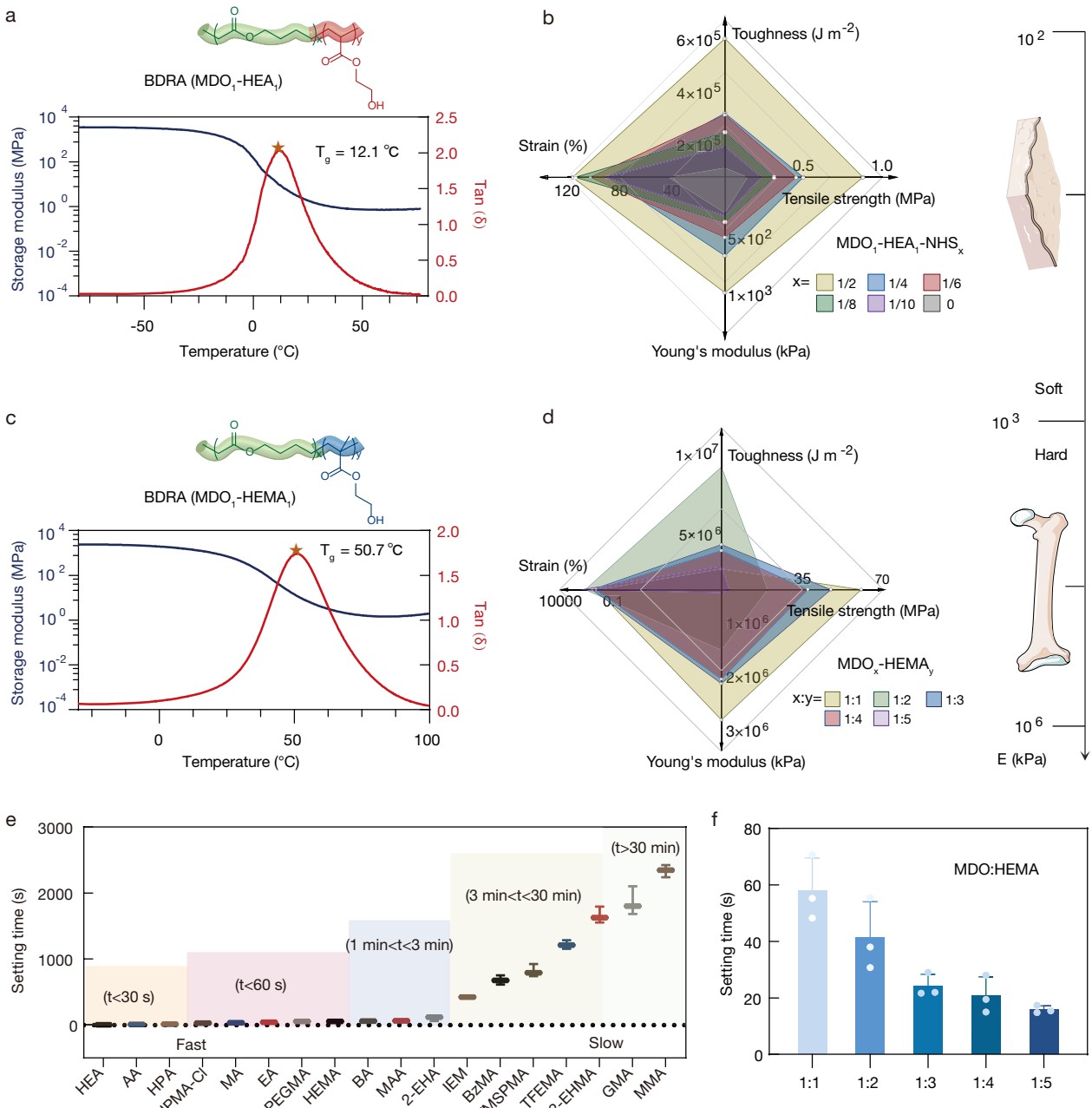

**Fig. 4 | Tunable mechanical properties and setting time of BDRAs. a**, **b** Glass transition temperature (**a**) and mechanical properties (elastic modulus, tensile strength, elongation at break, and toughness) with respect to polymer compositions for BDRAs (MDO-HEA) (**b**). **c**, **d** Glass transition temperature (**c**) and mechanical properties (elastic modulus, tensile strength, elongation at break, and toughness) with respect to polymer compositions for BDRAs (MDO-HEMA) (**d**). **e** Setting time of BDRAs, MDO with eighteen acrylate comonomers and a molar ratio of MDO to comonomer of 1:1. AA acrylic acid, HPA hydroxypropyl acrylate,

HPMA-Cl 3-chloro-2-hydroxypropyl methacrylate, MA methyl acrylate, EA ethyl acrylate, PEGMA poly (ethylene glycol) methacrylate, MAA methacrylic acid, 2-EHA 2-ethylhexyl acrylate, IEM 2-isocyanatoethyl methacrylate, BzMA benzyl methacrylate, TMSPMA 3-(trimethoxysilyl)propyl methacrylate, TFEMA trifluoroethyl methacrylate, 2-EHMA 2-ethylhexyl methacrylate, GMA glycidyl methacrylate, EMA ethyl methacrylate. Data are presented as the means ± SDs, $n = 3$ independent samples. **f** Setting time of MDO copolymerizing with HEMA at different monomer ratios. Data are presented as the means ± SDs, $n = 3$ independent samples.

monomers and hydrophilic acrylate monomers in a physiological environment. The adhesives showed strong bonding to various materials (including wet biological tissue and polymer-based substrates with low surface energy) due to their good wettability, high penetration, and formation of a covalent interpenetrating network. Combining the advantages of both ring-opening polymerization (intermittently introducing cleavable bonds within the backbone) and radical polymerization (easy synthesis, broad diversity of architectures, compositions, and

functionalities), the robust BDRA adhesives had both a biodegradable backbone with a controlled degradation rate and wide ranges of mechanical moduli (100 kPa–10 GPa) and setting time (seconds–hours).

The adhesives essentially precluded the instant polymerization of CA super adhesive for wet adhesion and achieved both in situ operation and in vivo degradation, which is different from traditional acrylate adhesives and biodegradable polyester adhesives. The adhesives outperformed sixteen well-recognized adhesives in terms of the

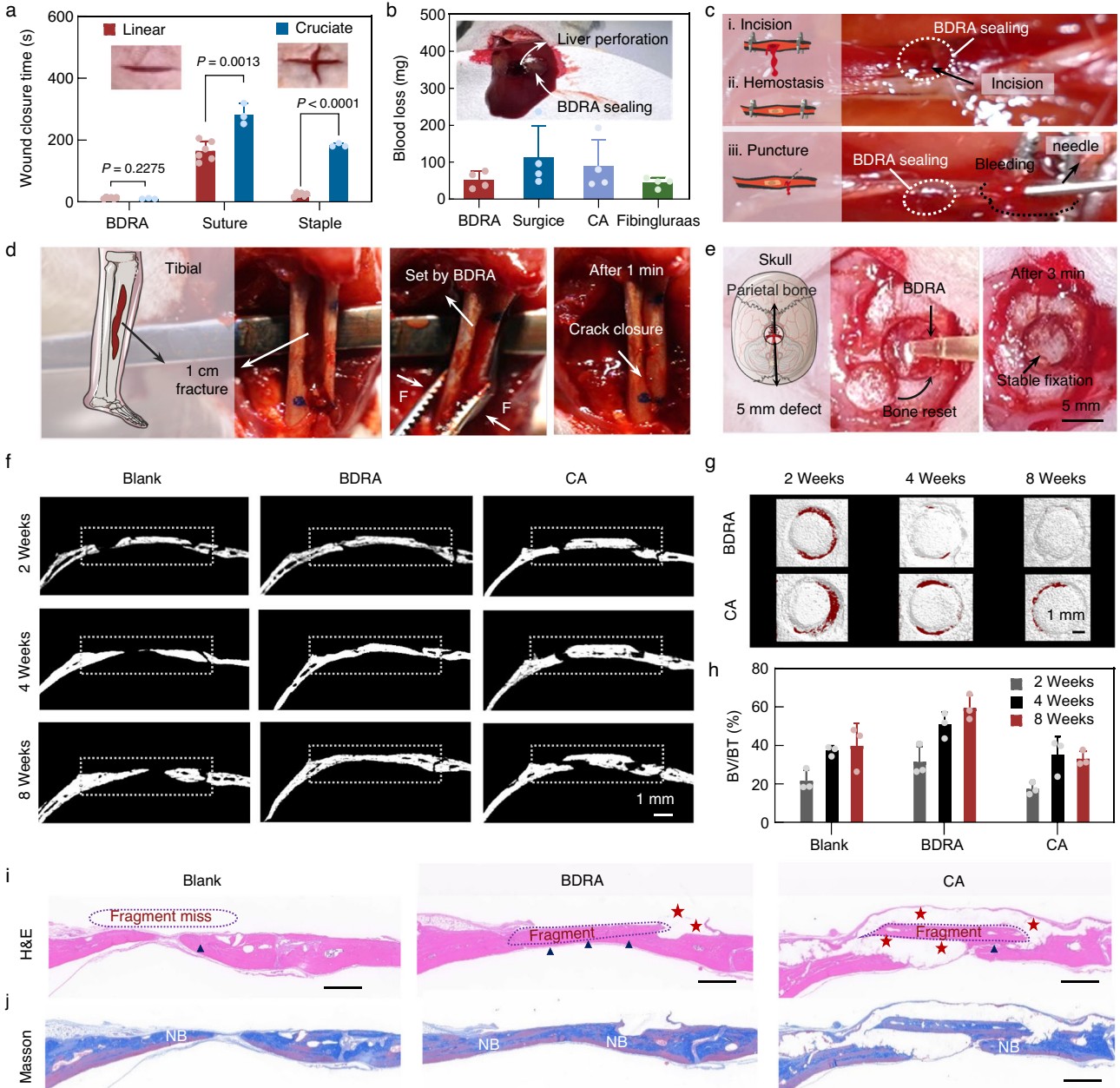

**Fig. 5 | Wound management, hemostasis, and in vivo adhesion of BDRAs.**
**a** Quantitative analysis of the implementation time of wound closure treatment by a BDRA (MDO$_1$-HEA$_1$-NHS$_{1/2}$) in a rat dorsal skin model with the treatment of sutures and staples as a control. Data are presented as the means ± SDs, $n$ = 6 independent samples for linear closure; $n$ = 3 independent samples for cruciate closure. **b** Weight of blood loss from 4-mm diameter perforated wounds in a liver injury model. Data are presented as the means ± SDs, $n$ = 4 independent samples. **c** Carotid artery sealing hemostasis by a BDRA (MDO$_1$-HEA$_1$-NHS$_{1/2}$) in a rat model. **d** In vivo experiments of semitruncated tibial fractures closed by a BDRA (MDO$_1$-HEMA$_1$). **e** Rapid fixation of the skull block with BDRA (MDO$_1$-HEMA$_1$) in an in vivo model of

the skull full-thickness defect. Scale bar, 5 mm. **f** Representative 2D Micro-CT images of rat skulls after fracture fixed by BDRA and Vetbond, and those without glue as a blank control. Scale bar, 1 mm. **g** Representative 3D CT reconstruction images of skull fracture. Scale bar, 1 mm. **h** Quantitative analysis of regenerated bone tissues by bone volume/total volume (BV/BT). Data are presented as the means ± SDs, $n$ = 3 independent samples. **i** Representative H&E staining for regenerated bone tissues. The red stars represent glues, and the blue triangles indicate the newly formed bone. Scale bar, 1 mm. **j** Representative Masson staining for regenerated bone tissues. NB represents new bone tissue. Scale bar, 1 mm. The experiments in (**i**, **j**) were repeated independently three times with similar results.

overall considerations of mechanical compliance, setting time, and degradability, so they offer new possibilities for personalized medicine ranging from tissue repair to implantable and wearable devices.

## Methods
### Materials and animals
All chemicals except BPO (Macklin, Shanghai, China) were obtained from Aladdin (Shanghai, China), and all were used without further

purification. Calcein-AM/PI live/dead staining assay kits and Cell Counting Kit-8 (CCK-8) were purchased from Solarbio (Beijing, China). The L929 murine fibroblast cell line and murine osteoblast precursor cells MC3T3-E1 were purchased from Shanghai FuHeng Biology Co., Ltd. PCL, Mw = 80,000, was produced by Perstorp (Malmö, Sweden). The engineering adhesive Scotch-Weld™ (epoxy adhesive DP100, cyanoacrylate adhesive PR100, urethane adhesive DP6310NS) and cyanoacrylate tissue adhesive Vetbond were purchased from 3 M

(Minnesota, USA). Baiyun adhesive was purchased from Guangzhou Baiyun Medical (Guangzhou, China). CoSeal surgical sealant was obtained from Baxter Healthcare Corporation (Massachusetts, USA). The fibrin sealant (Human) Fibingluraas was purchased from Shanghai BDRAS Blood Products Co., Ltd. (Shanghai, China). The 2-octyl cyanoacrylate adhesives Dermabond and Surgicel were purchased from Ethicon (Somerville, USA).

Animal tissues for ex vivo experiments were obtained from the local market or abattoirs. All animal experiments were carried out in compliance with and approved by the Animal Care and Ethics Committee of Changchun Institute of Applied Chemistry, Chinese Academy of Sciences. Female Sprague–Dawley rats (SD) and BALB/c mice (Animal Center of Jilin University, Changchun, China) were used for the in vivo experiments of this study. All rats were housed at an ambient temperature of 25 °C (24–26 °C) and humidity of 30% and allowed access to a standard diet and water ad libitum. BALB/c mice used for pulmonary toxicity tests were supplemented with fresh apples to provide water and nutrients.

## Preparation of the BDRAs
The BDRAs were prepared by the following procedure. MDO (48 μL, 0.42 mmol) and HEMA (52 μL, 0.42 mmol) monomers were mixed with the crosslinker ethylene dimethacrylate (EDMA, 2 μL, 0.0105 mmol) and oxidizing agent BPO (0.002 g, 0.0084 mmol) in a centrifuge tube to form the adhesive precursor. Then, the reducing agent DMPT (1.2 μL, 0.0084 mmol) was added to the mixture to prepare the desired $P(MDO_1-HEMA_1)$ adhesive for bone adhesion. The different feed molar ratios of MDO:HEMA in this study ranged from 1:1 to 1:5. $^1$H NMR and FT-IR were used to characterize the chemical structure of the copolymer of MDO and HEMA.

To form an adhesive for soft tissue adhesion, MDO (52 μL, 0.45 mmol) and HEA (48 μL, 0.45 mmol) were mixed with EDMA (0.85 μL, 0.00089 mmol) and BPO (0.002 g, 0.009 mmol) in a centrifuge tube to form the adhesive precursor, followed by adding DMPT (1.3 μL, 0.009 mmol) to afford the desired $P(MDO_1-HEA_1)$ adhesive. Additionally, acrylic acid N-hydroxysuccinimide ester (AAc-NHS) was added, and the different feed molar ratios of MDO:NHS in this study were varied from 1:0 to 1:2 to fabricate $P(MDO_1-HEA_1-NHS)$ adhesives.

## Characterization of BDRAs
Proton nuclear magnetic resonance ($^1$HNMR) spectroscopy of BDRA was recorded at room temperature in $CD_3OD$ on a Bruker 500 MHz spectrometer. The molecular weight and molecular weight distribution were determined by GPC in DMF, with the linear PMMA as a calibration standard.

## Mechanical tests
To measure the mechanical strength of the BDRAs, the samples were prepared and studied according to the standard tensile test methods (ASTM D638) using a universal testing machine (AGS-X, SHIMADZU, Kyoto, Japan). The samples were cut into a dumbbell shape with dimensions of $15 \times 2 \times 1\,mm^3$. All tests were performed by using a constant tensile speed of 50 mm min$^{-1}$. The stress–strain curves were recorded, and the tensile strength was established by dividing the maximum force by the cross-sectional area. The fracture toughness of the BDRA was calculated by integrating the area under the curves. Young's modulus was measured via the ratio of the stress value to its corresponding tensile strain value within the linear section of the plot (strain within 10%).

To measure the temperature dependence of the storage modulus ($E'$), loss modulus ($E''$), and loss factor ($\tan(\delta)$) of the BDRAs, the BDRAs were cut into a rectangular shape ($10 \times 6 \times 0.6\,mm^3$) and tested using a dynamic thermomechanical analyzer (DMA 850, TA Instruments, New Castle, USA). All tests were conducted with a frequency of 1 Hz and a preload force of 0.0001 N. The temperature ramp ranged with a rate of 3 °C/min from −80 °C to 80 °C and −30 °C to 180 °C separately for testing the viscoelasticity of the BDRAs (MDO-HEA) and the BDRAs (MDO-HEMA) with temperature. The curves of $E'$ and $\tan(\delta)$ (which represent $E''/E'$) with temperature were recorded. The glass transition temperature ($Tg$) was defined as the temperature corresponding to the maximum $\tan(\delta)$.

## Setting time and thermal effect of BDRAs
The setting time of the BDRAs was determined by the vial inversion method. In brief, 100 μL BDRA precursor was added to a 1.5 mL centrifuge tube and equilibrated at 37 °C for 30 min. Timing commenced with the addition of DMPT. The centrifuge tube was tilted every 5 s until the adhesive solution did not flow. The setting time and status were recorded. Each sample was measured at least 3 times. To elucidate the thermal effect, the adhesive was polymerized in situ on the pig skin, and the thermal images were collected by thermal imager (HT-19). The peak temperature was recorded and repeated at least 3 times for each sample. The acrylate homopolymer systems (PHEA and PHEMA) and commercially available CA adhesives (Vetbond) were used as references.

## Wettability of the adhesives
To test the surface wettability of the BDRAs, water contact angle measurements were performed using a drop-shape analyzer (DSA100, KRÜSS, Hamburg, Germany). Briefly, 2 μL water or BDRA precursors were dropped onto the surface of various substrates with a microsyringe. Then, the water contact angles were recorded. Each group of tests was repeated five times.

## Penetration of BDRAs
BDRA precursors with 2% w/w fluorescein isothiocyanate (FITC, excitation wavelength 490 nm) were prepared for the identification of BDRAs penetrating porcine skin. Briefly, after various BDRA monomers, EDMA, and BPO were mixed, FITC was dissolved in the mixture while stirring. To measure the diffusion depth of the BDRAs, 50 μL each of BDRA precursor without DMPT was placed on fresh porcine skin ($1.5 \times 1\,cm^2$). At certain time points, the adhesive-tissue hybrids were imaged by confocal laser scanning microscopy (CLSM700, Carl Zeiss, Jena, Germany) to measure the penetration depth of the adhesive over time. To determine the effect of the hydrophilic-hydrophobic composition on the penetration depth of the BDRAs, each BDRA precursor with DMPT was placed on the porcine skin or bone surface. After incubating the adhesive-tissue hybrids at room temperature for 24 h, confocal imaging and corresponding bright field images were collected and recorded.

## Adhesion force measurement
All adhesion forces were measured using a Universal Testing Machine (LR10K Plus, AMETEK -Lloyd, Florida, USA). For wet adhesion properties, the adherents were immersed in ultrapure water for 30 min before application to characterize the adhesion of the adhesives.

To characterize adhesive strengths between the BDRAs and hard tissue, bovine leg bones were cut into sheets with a width of 2.5 cm. To measure the shear strength, 60 μL adhesive was applied onto a bone sheet, and another sheet was bonded in a lap shear manner. After being set at room temperature for 30 min, the adhered bone pieces were subjected to a lap shear test. All tests were performed with a 2.5 kN sensor at a constant tensile speed of 5 mm min$^{-1}$. The shear strength was quantified by dividing the maximum force by the adhesive area.

For further tensile strength tests of the BDRAs to bone tissue, the bone was cut into regular bone blocks with an adhesion area of a 1 cm width and length for the bottom. Then, 30 μL adhesive was applied onto one bone block, and the other block was bonded in a butt joint manner. All tests were conducted using a 2.5 kN sensor at a constant tensile speed of 5 mm min$^{-1}$. The tensile strength was quantified by dividing the maximum force by the adhesive area.

For the 3-point bend test, the bone was cut into columns with a width of 10 mm and a length of 40 mm. Then, 30 μL of adhesive was applied onto a bone column, and another column was bonded in a butt joint manner. After setting for 30 min at room temperature, the adhered bone columns were subjected to a 3-point bend test. The vertical force with a testing speed of 0.6 mm min⁻¹ was applied until fracture. The displacement (deflection) and blend force curve were recorded. All tests were performed with a 500 N sensor. The specimen was tested flatwise on a support span ($L$ = 60 mm) by the standard test methods to measure the flexural properties (ASTM-D790-10). The flexural stress ($\sigma_f$) at any point was calculated based on the load–deflection curve by Eq. 1:

$$\sigma_f = 3PL/2bd^2 \qquad (1)$$

where $P$ is the load at a given point on the load–deflection curve, $L$ refers to the support span, $b$ is the width of the beam tested, and $d$ is the depth of the beam tested.

To quantify the adhesive force of BDRAs on soft tissue, fresh porcine tissues purchased from the local market were cut into a rectangular shape with a width of 2.5 cm without any pretreatment. All porcine tissues were sealed with plastic wrap before and after the test to prevent dehydration. Before applying adhesives to samples, the porcine tissues were wiped with 75% alcohol to remove excessive oil and impurities. All tests were conducted after 30 min of bonding at room temperature with a 500 N sensor. Except for the T-Peel test, the other tests were carried out at a speed of 50 mm min⁻¹.

To measure the shear strength, adhered porcine skins with an adhesive area of a width of 2.5 cm and length of 1 cm were tested by the standard shear test (ASTM F2255). A total of 60 μL adhesives were applied onto one piece of porcine tissue, and another piece of porcine tissue was bonded in a lap shear manner. The lap–shear curve was recorded until adhesive failure. The apparent shear strength was quantified by dividing the maximum adhesion force by the adhesion area.

For the tensile test of BDRA adhered to soft tissue, adhered porcine skin with an adhesive area of 2.5 cm width and 2.5 cm length was tested by the standard tensile test (ASTM F2258). In brief, 100 μL of BDRA was applied to the overlap area of the sample tissue. The back side of the samples was glued to the test fixture by cyanoacrylate adhesives before testing. The load force curve was recorded until adhesive failure. The tensile strength was quantified by dividing the maximum adhesion force by the bond area.

To characterize the interfacial toughness, adhered sample tissues with a width of 2.5 cm were prepared according to the standard peel test (ASTM F2256). In brief, 100 μL adhesive was applied onto tissue samples 3 cm in length. All tests were carried out at a constant peeling speed of 250 mm min⁻¹. The peeling force curve and plateau force in the steady state of the peeling process were recorded. The interfacial toughness was calculated by dividing the platform force by the width of the porcine skin.

For the wound closure test, porcine skin with a 2.5 cm width and 10 cm length was placed head-to-head on the table. A total of 50 μL adhesive was uniformly injected in and around the space between two tissue samples with adhesion areas of 2.5 cm width and 1 cm length. Then, the tissue samples were subjected to the standard test (ASTM F2458). The wound closure strength was determined by dividing the maximum load force by the adhesion area.

### In vitro and in vivo biodegradation

Before the degradation tests, BDRAs were added to a Teflon plate mold (diameter of 8 mm and thickness of 0.2 mm) for preforming. After setting for 24 h, the adhesives were weighed as $W_0$. Then, some of the BDRAs were soaked in vials containing 3 mL PBS and incubated at 37 °C for in vitro hydrolysis. The corresponding number of BDRAs were immersed in 3 mL of 1 M NaOH for accelerated degradation. At

each time interval, the samples were removed from the incubation medium. After being thoroughly washed with deionized water, the samples were freeze-dried and weighed as $W_d$. The in vivo biodegradation was characterized by subcutaneously implanting the BDRAs into rats. PHEMA and CA (Vetbond) were prepared and tested in the same way, along with PCL as control groups. At a given time point, the rats were anesthetized, and the samples were removed. After being washed with deionized water, the samples were freeze-dried and weighed ($W_d$). The percentage ratio of the residual amount of the sample was calculated using Eq. 2:

$$\text{Residual amount \%} = \frac{W_d}{W_0} \times 100\% \qquad (2)$$

### In vitro biocompatibility evaluation

To fully characterize the cytocompatibility of the BDRAs, both extract and direct contact tests were performed. In the extraction method, 50 mg adhesive was first incubated in 1 mL Dulbecco's modified Eagle's medium containing 2% v/v fetal bovine serum (2% DMEM) at 37 °C for 24 h to prepare the conditioned medium. Pristine 2% DMEM was utilized as a control group. Then, the cultured L929 or MC3T3 ($1 \times 10^4$ cells/well) cells in 96-well plates were treated with the conditioned medium and incubated at 37 °C for 24 h with 5% $CO_2$. Cell viability was detected via a CCK-8 assay kit (Beyotime, Shanghai, China) by changing the medium to 100 μL fresh culture medium with 10% v/v CCK-8 and culturing for 2 h at 37 °C with 5% $CO_2$. The absorbance of the incubated solution was determined at 450 nm by a microplate reader (TECAN SUNRISE, Maennedorf, Switzerland).

To test the cytotoxicity of the BDRAs by the direct contact method, the adhesives were polymerized in situ at the bottom of 96-well plates for 24 h. After being washed with PBS twice, L929 cells ($1 \times 10^4$ cells/well) were seeded in 96-well plates and co-incubated at 37 °C for 24 h with 5% $CO_2$. Then, the direct contact toxicity was quantified by CCK-8 assay.

To further identify the biocompatibility of the BDRAs in vitro, the live/dead assay was carried out with a live/dead cytotoxicity kit according to the manufacturer's instructions. Qualitative images of live cells were obtained using CLSM700.

### In vivo biocompatibility

The in vivo biocompatibility of the BDRAs was measured by subcutaneous implantation in the abdomen of female SD rats weighing 100–120 g. Before implantation, the BDRAs were prefabricated into a circular shape with a diameter of 8 mm and a thickness of 0.2 mm. To implant in the space of the abdomen subcutaneously, the back hairs were removed from the anesthetized rats, and a 1 cm incision was created in the center of the rat's back. Then, the adhesive samples were placed in the space between the back skin and muscle fascia.

Also, the BDRAs were injected and in situ polymerization in the abdominal subcutaneous pockets. Commercial cyanoacrylate adhesives Baiyun and Vetbond were used as control groups. All incisions were immediately sterilized and sutured. At certain time points, the rats were euthanized, and the regions of interest were collected and fixed with tissue fixative (Servicebio, Wuhan, China) for histological staining and analysis.

### Wound closure model

To test the effect of wound closure by the BDRAs, a total of 12 female SD rats weighing 140–160 g were randomly divided into two groups for linear and cruciate incision. Rats were anesthetized with 5% v/v isoflurane and maintained anesthesia on the operating table with 1–1.5% v/v isoflurane. Four line or cruciate incisions (1 cm/line) on the dorsal skin were created on the rats. Then, BDRAs, 3-0 PGA interrupted sutures (R315, Jinhuan, Shanghai, China), staples, and Vetbond were

each applied to close the wounds. Images were acquired at certain time points, and tissues below the wound were collected for further histological analysis.

## Hemostatic sealing of small-mammal models

To demonstrate the rapid sealing hemostasis of the BDRAs, hemostasis models of liver perforation and carotid and caudal vein injuries were conducted. Commercial products of Vetbond, Fibingluraas, and Surgical were used as controls. Female SD rats weighing 140–160 g were randomly divided into four groups ($n = 4$) and were used for all hemostatic experiments. Before the tests, the rats were anesthetized with 5% v/v isoflurane and maintained anesthesia on the operating table with 1–1.5% v/v isoflurane.

For liver hemostasis and healing, the left lateral lobe of the liver was exposed, and a 4-mm-diameter defect was created with a biopsy punch. Each BDRA ($MDO_1\text{-}HEA_1\text{-}NHS_{1/2}$) and commercial adhesive sample was applied to the defects. The hemostasis properties of various adhesive samples were observed until the wound did not bleed. The bleeding time and quantity were measured and recorded. Then, the damaged liver tissues were collected after 1, 7, and 14 days for further H&E staining and immunohistochemical imaging of IL-6 and TNFα.

For artery-sealing hemostasis, the right carotid artery of rats was separated by blunt dissection and clamped at the distal and proximal ends. Then, an incision on the vessel was made with a 26 G needle. A BDRA ($MDO_1\text{-}HEA_1\text{-}NHS_{1/2}$) was used to seal the wound. The vascular clamps were released after 3 min, and the hemostatic effect was observed.

For rat tail sealing hemostasis, an 8-mm incision was cut to destroy the caudal vein on the tail that was 2 cm away from the root of the rat tail. After slightly wiping the bleeding surface with medical gauze, each BDRA ($MDO_1\text{-}HEA_1\text{-}NHS_{1/2}$) and commercial adhesive sample was applied. The injured tail was moved regularly on pre-weighed filter paper until the wound did not bleed. Blood loss was quantified and recorded.

## Histological analysis

After all tissue samples with regions of interest were collected and fixed in 4% w/v formalin, they were used for histological analyses. Immunostaining and imaging were both completed by Wuhan Servicebio Technology Co., Ltd.

## Bone damage and healing model

For the skull defect model, female SD rats were randomly divided into three groups ($n = 3$) at each time point. The rats were anesthetized, and the hair of the head was removed. After being sterilized with 2% w/v iodophor cleaning, the top of the skulls was exposed, and a 5-mm diameter full-thickness bone defect was created using a dental ring drill (5-mm outer diameter). Then, the round skull fragment was dissected with a periosteal stripper, and 25 μL of Vetbond and BDRA ($MDO_1\text{-}HEMA_1$) was injected into the defect. The skull fragment was immediately replaced, and soft tissues were closed with sutures. After 2, 4, and 8 weeks, the rats were euthanized, and the skulls of the rats were collected and fixed with 4% paraformaldehyde. All samples were scanned by micro-CT (SkyScan1172, Belgium) at 80 kV, 100 μA, and reconstructed and analyzed. The samples of rat skulls were subsequently decalcified and stained for H&E and Masson staining.

For the tibial fracture model, the hair on the calf of rats was removed and sterilized. Then, a longitudinal incision was created to expose the tibial diaphysis. A 5-mm semi-truncated defect was observed to simulate the tibial fracture. Then, BDRA ($MDO_1\text{-}HEMA_1$) was applied to splice the bone fragments of the fracture.

## Statistical analysis

The data are displayed as the means ± standard deviations (SDs). All statistical analyses were conducted using GraphPad Prism software (GraphPad Software Inc., USA) from at least three parallel experiments. One-way analyses of variance (ANOVA) followed by Tukey's post hoc test were used for the statistical analysis between multiple groups, and two-sample Student's $t$-test was used for comparisons between two samples.

## Reporting summary

Further information on research design is available in the Nature Portfolio Reporting Summary linked to this article.

## Data availability

All data are available in the main text, Supplementary Information, or Source Data file. Source data are provided in this paper. If any raw data files are needed in another format, they are available from the corresponding author upon request. Source data are provided in this paper.

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

## Acknowledgements

We thank Prof. Q. Zhao (from the Key Laboratory of Material Chemistry for Energy Conversion and Storage at Huazhong University of Science and Technology) for useful discussions and suggestions. This work is supported by the National Key Research and Development Program of China (No. 2020YFC1106900 to S.L.). We acknowledge the High-Tech Research & Development Program of CAS-WEGO Group for support.

## Author contributions

R.Y., B.C., and S.L. conceived the idea and designed the research. R.Y. and B.C. fabricated and characterized the materials. R.Y. and X.Z. developed the methods and performed the in vitro, ex vivo, and in vivo studies. R.Y., X.Z., and Q.Y. contributed to the data analysis. J.Y. and S.L. supervised the study. R.Y. wrote the paper with input from all authors. S.L., Q.Y., and B.C. edited the original draft.

## Competing interests

R.Y., X.Z., B.C., Q.Y., and S.L. are inventors on a patent application for the BDRA adhesive reported in this paper (US patent publication No. US-2023-0211043-A1). The remaining authors declare no competing interests.
