## [Peer Review File · Nature Communications]

Reviewers' Comments:

Reviewer #1:

Remarks to the Author:

This work reports the design and performance of new polymeric adhesives capable of forming strong adhesion on diverse surfaces. The new adhesives feature radical ring-opening polymerization, allowing for fine-tuning of the gelation speed and degradation profiles. The material system is novel and interesting, supported by extensive research efforts and characterizations. The proposed mechanism is convincing as well as the presented data. The performance demonstrated by the new adhesives is impressive in terms of adhesion strength and tunability. The manuscript is well written and the figures are well crafted. Overall, the reviewer is happy with the manuscript and endorses its publication in NC after the authors address the following concerns.

1. Please comment the thermal effects of the adhesives. A recognized limitation of polymeric adhesives is substantial heat released from the polymerization. It is a known issue with cyanoacrylate. Given the high concentration, the proposed system might have similar effects.
2. How does the adhesion energy achieved by the new adhesives compare with other adhesives? The reported value seems much lower than tough adhesives.
3. Programmability often refers to the tunability within a single material system. It seems overstatement here because the users have to modify the chemistry for different adhesion/degradation/properties.
4. Cytocompatibility. Compared to hydrogel adhesives, the adhesives induce some cytotoxicity. So the authors should not claim noncytotoxic in the text on page 7.
5. Animal experiments. It should be cruciate incision, not crucial incision. The discussion and experimental section on animal experiments are considered weak in this manuscript compared to other parts. Please provide more data to support the in vivo performance, for instance, histology and staining.

Reviewer #2:

Remarks to the Author:

Luan et al. reported the preparation of a backbone-degradable adhesive via in-situ radical ring-opening polymerization of CKA and (meth)acrylate monomers. They described that the use of hydrophilic vinyl monomers allows for effective penetration into substrates before the polymerization, hence increasing the adhesion strength of polymers. Their copolymers are degradable under basic and physiological conditions.

In terms of originality, I'm not sure the work represents a significant advance. Agarwal et al. have previously shown the structure of poly(MDO-co-methacrylate) by in-situ cross-linking of polymer chains to form an enzymatically degradable adhesive (DOI: 10.1021/acsbiomaterials.5b00217), while Luan and coworkers have improved the adhesion of a similar structure by penetrating the precursors into substrates before making the polymers.

I would therefore recommend publication in a more specialized journal focused on bio-applications

There are some questions and comments:

1. It is quite surprising that the copolymer compositions are very similar to the feed compositions (Figure S1) since the reactivity ratio of MDO monomer is much smaller than that of (meth)acrylate monomer. Are the polymerization conversions complete? Is there any degradation or hydrolysis of MDO during the polymerization? Do the authors have any explanation?
2. Macromolecular characteristics (M_n , M_w/M_n) of the different copolymers (see Figure S1) are absent.

3. Cytotoxicity studies of the adhesive precursors (monomers, oxidizing agent, and crosslinking agent) should be conducted as they penetrate and diffuse in the tissues. The polymerization conversions are likely incomplete due to the later addition of a reducing agent, which results in residual precursors remaining in the tissues or organs. Evaluating the cytotoxicity of precursors is important since it distinguishes the adhesive from cyanoacrylate superglue, as authors have highlighted in the introduction."

Reviewer #3:

Remarks to the Author:

Current study has put forward the design of a programmed backbone-degradable robust adhesives via radical ring opening polymerization. The adhesive was claimed to exhibit superior adhesive strength toward diverse materials and tissues as well as tunable setting time and degradation rate. Tissue adhesive has been attracting attention due to its huge potential in clinical medicine. The adhesive system from Yang et al. has been highlighted on its capacity of promoting the penetration of adhesive molecule into substrate to achieve higher adhesion strength and allowing programmed degradability. These points could provide valuable clues for researches in tissue adhesive materials. However, based on current results, the performance of designed adhesive was not comprehensively characterized and could hardly satisfy the high demand of Nature Communication on the aspect of innovation. Overall, I suggest a rejection for this manuscript.

Associated problems have been listed as follow, hopefully to help you improve your manuscript.

1. Ring-opening polymerization has been employed in construction of tissue adhesives. The innovation of current one should be discussed compared with previous ones.
2. The specific usage of designed adhesive was ambiguous in this study. No specific target tissue was mentioned. Despite that bone, skin, artery, liver and heart tissues were employed for assessment of adhesiveness, corresponding characterization methods remained on relatively superficial level.
3. The degradation performance of cyanoacrylate adhesive should also be studied for comparison with BDRAs.
4. In vivo investigation in this study could hardly fully characterize the performance of designed adhesive. The degradation of adhesive in vivo was accompanied by host tissue ingrowth and penetration. In vivo characterization of a degradable adhesive materials should include the observation of adhesive-tissue interaction at different time points after implantation.

Point-to-point response to reviewers

Reviewer #1:

Comments:

This work reports the design and performance of new polymeric adhesives capable of forming strong adhesion on diverse surfaces. The new adhesives feature radical ring-opening polymerization, allowing for fine-tuning of the gelation speed and degradation profiles. The material system is novel and interesting, supported by extensive research efforts and characterizations. The proposed mechanism is convincing as well as the presented data. The performance demonstrated by the new adhesives is impressive in term of adhesion strength and tunability. The manuscript is well written and the figures are well crafted. Overall, the reviewer is happy with the manuscript and endorses its publication in NC after the authors address the following concerns.

Response:

We genuinely appreciate the positive comments on our manuscript. According to these comments, we have revised our manuscript totally.

Question 1: Please comment the thermal effects of the adhesives. A recognized limitation of polymeric adhesives is substantial heat released from the polymerization. It is a known issue with cyanoacrylate. Given the high concentration, the proposed system might have similar effects.

Response:

Thanks for the reviewer's advice. To evaluate the thermal effect of the BDRA family, the adhesive was polymerized *in-situ* on the pig skin, and the temperature change was monitored by a thermal imager (HT-19). The acrylate homopolymer systems (PHEA and PHEMA) and commercially available cyanoacrylate adhesives (Vetbond) were used as references. As shown in Figure R1, the tested temperatures of *in-situ* polymerization of BDRAs (MDO-HEMA, < 45 °C) were below the thresholds for bone necrosis (56 °C) (*J Orthop Res*, 2023, 41, 614.), and they were also lower than the Vetbond (~54 °C) and PHEMA references. Notably, these temperatures are much lower than that of the *in-situ* polymerization of PMMA (100-101 °C), which is a main component of commercial bone cement (*Technol Health Care*, 2012, 20, 337.).

The tested temperatures of *in-situ* polymerization of BDRAs (MDO-HEA) ranging from 39 to 59 °C, are much lower than those for PHEA (~100 °C). Furthermore, the aforementioned temperature decreased with an increase in the proportion of MDO due to its lower activity compared to acrylate monomers.

The results and methods have been listed in the revised manuscript and Supplementary Information (Supplementary Fig. 4) marked in yellow. The relevant

details in main text (page 4, line 111-117) as follows: “The thermal effect of BDRAs was evaluated during *in-situ* curing on pig skin (Supplementary Fig. 4). The tested temperatures lower than 45 °C were observed for the BDRA (MDO-HEMA) family, which were lower than that of Vetbond (~54 °C) and the thresholds for bone necrosis (56 °C)⁴¹. And the tested temperatures of the BDRA (MDO-HEA) family ranging from 39 °C and 59 °C were much lower than that of the acrylate homopolymer system (~100 °C), suggesting a lower damage probability of skin⁴².”

Figure R1. Thermal effects of BDRAs on porcine skin. **(a)** The peak temperature of BDRAs and Vetbond and acrylate homopolymer system (PHEA and PHEMA) by applying 20 μ L of adhesive to an area of 1cm \times 1cm on the surface of the porcine skin. Data are presented as means \pm SDs, n = 3-6. **(b)** Representative thermal images of the highest temperature on the surface of pigskin captured by a thermal imager.

Question 2: How does the adhesion energy achieved by the new adhesives compare with other adhesives? The reported value seems much lower than tough adhesives.

Response:

In this work, BDRAs are designed as a universal strategy applicable to both hard and soft tissues adhesion. For hard tissues such as bone with a high modulus and low

strain, they often require high bonding strength to obtain a solid adhesion. In this scenario, the adhesion strength is the most important parameter and is given priority consideration. The adhesion strength of BDRAs on wet bone is superior to that of seven well-recognized bone adhesives (Fig. 1f). In our reference experiments, the adhesion strength of BDRAs on wet bone was >16 MPa by the standard flexural test (~4 MPa for CA references, ~0.2 MPa for Coseal, ~0.1 MPa for Fibingluraas) (Fig. 1h).

For soft tissue such as pigskin, the adhesion strength of BDRAs on pigskin is superior to that of nine previously-reported adhesives. As for one of the most important parameters, adhesion energy, BDRAs to soft tissue pigskin is ~200 J m⁻², which is higher than that of commercial adhesives including octyl cyanoacrylate (~20 J m⁻² for Baiyun, ~57 J m⁻² for Dermabond), the fibrin-based Fibingluraas (~38 J m⁻²) and PEG-based COSEAL (~56 J m⁻²) (seen Supplementary Fig. 2). However, the adhesion energy of BDRAs is much lower than the recently-reported excellent tough hydrogel adhesives (>1000 J m⁻², see Science, 2017, 357, 378; Science 2022, 377, 751.).

Under the guidance of the above tough hydrogel adhesives, we will further improve the adhesive energy of BDRAs for specific applications of soft tissue.

Question 3: Programmability often refers to the tunability within a single material system. It seems overstatement here because the users have to modify the chemistry for different adhesion/degradation/properties.

Response:

In our work, the “programmable” was originally used to show the structure and performance of the BDRA family can be designed and regulated according to requirements. Based on the reviewer’s suggestion (Programmability often refers to the tunability within a single material system), we replaced the homologous words of programmability including programmed and programmable with **tunable** in the revised manuscript marked in yellow.

Question 4: Cytocompatibility. Compared to hydrogel adhesives, the adhesives induce some cytotoxicity. So the authors should not claim noncytotoxic in the text on page 7.

Response:

For the accuracy of the statement, we have revised noncytotoxic as **low cytotoxicity** in the revised manuscript and marked in yellow. The correct (page 8, line 216-218) as follows: “Additionally, the implanted BDRAs showed lower cytotoxicity (with cell viabilities >87%) than Vetbond (<60 %) towards osteoblast precursor MC-3T3 cells and mouse fibroblast L929 cells (Fig. 3f and Supplementary Fig. 11).”

Question 5: Animal experiments. It should be cruciate incision, not crucial incision. The discussion and experimental section on animal experiments are considered weak in this manuscript compared to other parts. Please provide more data to support the *in vivo* performance, for instance, histology and staining.

Response:

Thanks for your suggestion, we have corrected the wording of the cruciate incision in the revised manuscript (pages 11, line 287 and pages 17, line 548).

According to the reviewer's suggestion, we have conducted in-depth studies of the BDRA (MDO-HEA) on the liver perforation model (new Supplementary Fig. 26) and the BDRA (MDO-HEMA) on the skull fracture model (new Fig. 5) to support the *in vivo* performance for soft tissue and hard tissue, respectively.

More histological and staining data, such as H&E and Masson staining (new Figs. 5i-j, Supplementary Figs. 12 and 13), immunostaining of lymphocyte (CD3) and macrophages (CD68) (new Figs. 3h and 3i), and immunohistochemical imaging of inflammatory cytokines [IL-6 and tumor necrosis factor- α (TNF- α)] (new Supplementary Figs. 25d and 25e), were supplemented in the revised manuscript. And the micro-CT images of bone fracture were also provided (new Figs. 5f-h).

The relevant details have been supplemented and highlighted in yellow in the revised manuscript (pages 8 and 11) and Supplementary Information (pages 13, 14, 26, and 27).

Thank you again!

Reviewer #2:

Comments:

Luan et al. reported the preparation of a backbone-degradable adhesive via in-situ radical ring-opening polymerization of CKA and (meth)acrylate monomers. They described that the use of hydrophilic vinyl monomers allows for effective penetration into substrates before the polymerization, hence increasing the adhesion strength of polymers. Their copolymers are degradable under basic and physiological conditions. In terms of originality, I'm not sure the work represents a significant advance. Agarwal et al. have previously shown the structure of poly(MDO-co-methacrylate) by in-situ cross-linking of polymer chains to form an enzymatically degradable adhesive (DOI: 10.1021/acsbiomaterials.5b00217), while Luan and coworkers have improved the adhesion of a similar structure by penetrating the precursors into substrates before making the polymers. I would therefore recommend publication in a more specialized journal focused on bio-applications

Response:

Thanks for the reviewer's positive comment. We regret not clearly demonstrating the uniqueness and innovation of our work in the original manuscript.

Biodegradable strong adhesives with *in-situ* rapid adhesion are significantly important in technical and clinical applications. One of the primary challenges is achieving the strong adhesion *in-situ* under physiological environments (wet, room temperature, oxygen/air), and the adhesion strength (~kPa) of most reported adhesives is still unsatisfied for many fields, such as bone adhesion (which requires ~MPa of adhesion strength) [see Nature, 2019, 575, 169]. An additional challenge is to achieve biodegradability without compromising adhesion strength. Besides strong adhesion to tissue, an ideal tissue adhesive must be biodegradable to achieving initial mechanical integration and subsequent bio-integration, avoiding the secondary surgical removal. However, most strong adhesives are nondegradable. Nature-derived adhesives are generally biodegradable, but their adhesion strengths are one order of magnitude weaker than synthetic adhesives due to their intrinsic low cohesive strength [see Science, 2017, 358, 872-873].

Our work firstly proposed a universal strategy to prepare a backbone-degradable robust adhesive (BDRA) by *in-situ* radical ring-opening polymerization (rROP). The BDRAs exhibited strong adhesion to diverse materials even for wet biological tissue (e.g., wet bone >16 MPa, superior to the commercial cyanoacrylate (CA) super-adhesive ~4 MPa), and outperformed sixteen well-recognized adhesives including non-degradable category. The BDRAs avoid the drawbacks of cyanoacrylate super-adhesives in wet adhesion, and achieve good *in vivo* biodegradation, which different from traditional acrylate adhesives and biodegradable polyester adhesives. This work will pioneer the *in-situ* rROP strategy for preparing adhesives, and transformative advances in the tunable synthesis of degradable robust adhesives will probably be comparable to the invention of CA super-adhesive.

Specifically, our work is essentially difference from the work by Agarwal group in the adhesion mechanism, preparation methods, performances and application scenario:

(1) Adhesion mechanism

In our work, the hydrophobic cyclic ketene acetal (CKA) and hydrophilic acrylate comonomer mixture of the BDRA precursor allows it to effectively wet and penetrate substrates, subsequently form a deep, high modulus, covalently interpenetrating network with adherent tissues via redox-initiated *in situ* rROP, so as to obtain strong adhesion (**the adhesion strength up to ~ 10 MPa level**) (Figure R2 a). In contrast, in the work of Agarwal group, rROP was used to pre-prepare degraded polymers, followed by functionalized with the catechol-groups, which achieves the DOPA-mimetic adhesion (**the adhesion strength up to ~10 kPa level**) (Figure R2 b). **Due to**

the fundamental difference of adhesion mechanisms between the two works, their difference in adhesion strength can reach up to three orders of magnitude.

Figure R2. The adhesion mechanism and adhesion strength of (a) BDRAs in our work and (b) adhesives in work of Agarwal et al.

(2) Preparation methods

In our work, BDRAs consist of degradable polymer were facilely prepared through *in-situ* rROP one-step process initiated by redox at physiological temperature for several seconds to hours (Figure R3 a). In contrast, in the work of Agarwal group, the degradable polymers were first prepared by rROP at 60 °C for 2 hours, followed by precipitating in hexane and drying in vacuum at 60 °C for 48 hours (Figure R3 b). And the obtained degradable polymers were modified with the catechol-groups at 65 °C for 20 hours under argon, and finally achieving the adhesion strength through H₂O₂, Fe³⁺ oxidation crosslinking. **This work involves in multi-step process for more than 70 hours, heating, organic solvent, vacuum, and gas protection; In contrast, our work just involves in one-step process for several seconds to hours without any solvents and gas protection.**

Figure R3. (a) The schematic of adhesion process in our work. (b) The synthesis of poly(MDO-co-GMA-co-OEGMA) and catechol-group functionalization in work of Agarwal et al.

(3) Materials performances and applications

The BDRAs through *in-situ* rROP present good universality, and the eighteen acrylate comonomers have been tested to confirm the feasibility in our work (Figure R4). By adjusting the type of acrylate monomers, the BDRAs can be prepared with different adhesion strength, bulk modulus, setting time and degradability, so they can be applied to the wide application scenarios from hard tissue to soft tissue adhesion (e.g., instant haemostasis and fracture adhesion fixation). In contrast, the *ex-situ* polymerized adhesive through pre-polymerization and post-modification in the work of Agarwal group don't provide adjustable data. Just for the curing time of the adhesive, it can be boldly inferred that its setting time windows should be similar, thus its adaptability to medical scenarios should be limited.

Comonomer name	CAS number	Setting time (t)	Structural formula	Classification
2-Hydroxyethyl acrylate (HEA)	818-61-1	5 s		(t < 30 s)
Acrylic acid (AA)	79-10-7	9 s		
Hydroxypropyl acrylate (HPA)	25584-83-2	12 s		
3-chloro-2-hydroxypropyl methacrylate (HPMA-Cl)	13159-52-9	31 s		(t < 60 s)
Methyl acrylate (MA)	96-33-3	32 s		
Ethyl acrylate (EA)	140-88-5	45 s		
Poly (ethylene glycol) methacrylate (PEGMA)	25736-86-1	55 s		
2-Hydroxyethyl methacrylate (HEMA)	868-77-9	58 s		(1 < t < 3 min)
Butyl Acrylate (BA)	141-32-2	60 s		
Methacrylic acid (MAA)	79-41-4	64 s		
2-Ethylhexyl acrylate (2-EHA)	103-11-7	2 min		(3 < t < 30 min)
2-Isocyanatoethyl methacrylate (IEM)	30674-80-7	7 min		
Benzyl methacrylate (BzMA)	2495-37-6	12 min		
3-(Trimethoxysilyl)propyl Methacrylate (MPTMS)	2530-85-0	13 min		(t > 30 min)
2,2,2-Trifluoroethyl methacrylate (TFEMA)	352-87-4	20 min		
2-Ethylhexyl methacrylate (2-EHMA)	688-84-6	28 min		
Glycidyl methacrylate (GMA)	106-91-2	31 min		(t > 30 min)
Methyl methacrylate (MMA)	80-62-6	40 min		

Figure R4. MDO copolymerizing with eighteen comonomers initiated by redox under environmental conditions.

In summary, we prepared a backbone-degradable robust adhesive (BDRA) by utilizing the *in-situ* radical ring-opening polymerization (rROP) for the first time in this manuscript. The BDRA was demonstrated with high adhesion strength for different tissues and the tunable performances, including the mechanical performances, biodegradability *in vivo*, and *in situ* setting time. It is effective to solve the contradictory problem of achieving the strong adhesion and biodegradability simultaneously in the fabrication of tissue adhesives. Meanwhile, the adhesion mechanism of BDRA was elucidated in detail as well in our work. All these have not

been reported previously. The new robust adhesive we proposed in this work may provide inspiration and reference for the design of tissue adhesives.

We have reorganized the manuscript and emphasized the novelty, as reflected in the title, Introduction (page 2, lines 52-69), and the end of text (page 13, lines 342-351) of the revised version.

Question 1:

It is quite surprising that the copolymer compositions are very similar to the feed compositions (Figure S1) since the reactivity ratio of MDO monomer is much smaller than that of (meth) acrylate monomer. Are the polymerization conversions complete? Is there any degradation or hydrolysis of MDO during the polymerization? Do the authors have any explanation?

Response:

Thanks for the reviewer's comments. The copolymer compositions of P(MDO-HEMA) with different ratio of MDO:HEMA were detected and quantified by the ^1H NMR tests, and we chose the characteristic H located on MDO and HEMA respectively to calculate the copolymer composition, and the results are shown as below in Figure R5. The reactivity ratio of MDO is smaller than that of meth acrylates monomer HEMA, so we can see that the copolymer composition (0.77:1) of P(MDO₁-HEMA₁) is less than that feed composition (1:1). With the increase of the amount of HEMA (from 1:2 to 1:5), the copolymer compositions are closed (or slightly below) to the feed compositions, we believe that this is because the MDO can be better inserted into the molecular chain to participate in the copolymerization with HEMA when the amount of HEMA is higher than that of MDO, from the point of view of chemical equilibrium.

For the radical polymerization, the polymerization conversion cannot be the 100%, so the residue of monomers is unavoidable. In addition, the MDO and HEMA in this work occur the bulk polymerization without solvent and water, we believe that there was no degradation or hydrolysis of MDO during the polymerization.

Figure R5. (a) The ^1H NMR spectrum of P(MDO-HEMA) with different ratio of MDO:HEMA, and the characteristic H located on MDO and HEMA respectively. (b) The copolymer composition of P(MDO-HEMA) calculated by the integral of characteristic H located on MDO and HEMA respectively according to the ^1H NMR spectrum.

Question 2:

Macromolecular characteristics (M_n , M_w/M_n) of the different copolymers (see Figure S1) are absent.

Response:

Following the reviewer's suggestion, we supplemented the information of macromolecular characteristics of different copolymers as shown in Figure R6. It should be noted that the M_n of copolymers P(MDO₁-HEMA₂), P(MDO₁-HEMA₃), P(MDO₁-HEMA₄) and P(MDO₁-HEMA₅) are less than that of P(MDO₁-HEMA₁). It is because that the polarity of copolymer is enhanced with the increase of HEMA content, which will cause the copolymer to adsorb the chromatographic column of GPC, resulting in the decrease of measured value of M_n . Even though we have screened the suitable solvent DMF as the eluent of GPC. The relevant information of

macromolecular characteristics of different copolymers have been supplemented in the Supplementary Fig. 1c of the revised Supplementary Information.

	Feed Composition MDO:HEMA	Mn (Da)	Mw/Mn
MDO ₁ -HEMA ₁	1 : 1	20187	2.84
MDO ₁ -HEMA ₂	1 : 2	11874	2.4
MDO ₁ -HEMA ₃	1 : 3	11438	2.39
MDO ₁ -HEMA ₄	1 : 4	12312	2.29
MDO ₁ -HEMA ₅	1 : 5	14017	2.33

Figure R6. The number average molecular weight (Mn) and distribution (Mw/Mn) of P(MDO-HEMA) were determined by GPC in DMF.

Question 3: Cytotoxicity studies of the adhesive precursors (monomers, oxidizing agent, and crosslinking agent) should be conducted as they penetrate and diffuse in the tissues. The polymerization conversions are likely incomplete due to the later addition of a reducing agent, which results in residual precursors remaining in the tissues or organs. Evaluating the cytotoxicity of precursors is important since it distinguishes the adhesive from cyanoacrylate superglue, as authors have highlighted in the introduction."

Response:

Thank you for your thoughtful consideration. To evaluate the cytotoxicity of residual precursors from BDRA adhesive systems, the *in-situ* cured BDRAs were immersed in tetrahydrofuran for 3 days, and the eluted monomers (MDO, HEMA, HEA, NHS, EDMA, BPO, DMPT) from the BDRA were determined using high-performance liquid chromatography (HPLC). The highest concentrations of residual MDO, HEMA, HEA, and NHS from adhesive systems were 2.4 mmol/L, 3.72 mmol/L, 1.45 mmol/L and 15.08 mmol/L, respectively. Although the eluted residual amounts of BPO and DMPT cannot be obtained due to overlapping peaks in HPLC, there was no potential cytotoxicity of BPO/DMPT in the formula of BDRAs because the cell viability was greater than 70% even if the BPO/DMPT remains completely, according to International Standard ISO 10993-5:2009 (Figure. R7a). In addition, the concentrations of residual MDO, HEMA, and NHS have the cell viability (>70%) significantly higher than that of the cyanoacrylate group, confirming the much lower potential cytotoxicity. (Figure. R7b)

Figure. R7 Biocompatibility of the residual precursors from BDRAs. (a) Cell viability of L929 co-incubated with involved precursors of BDRAs measured by CCK-8 assay. The theoretical maximum residual concentration of monomer, which does not participate in the reaction at all, is represented by 1, and is diluted ten times to 1/10000. CA was taken as control, data are presented as means \pm SDs, $n = 3-5$. (b) The concentrations of eluted residual precursors from BDRA systems and the corresponding cell viability. Representative H&E staining after BDRA in-situ curing of abdominal subcutaneous tissues at 1 and 2 weeks (c) and skin wound closure at 3 and 7 days (d). (e) The representative H&E staining of the heart, liver, spleen, lung, and kidney after the BDRAs implanted for 1 and 2 weeks, scale bar = 20 μm .

We observed that the highest concentrations of residual HEA (accumulated concentration after three days of soaking) appeared cytotoxic *in vitro* cell experiment. However, in the practical application scenarios, the actual concentration *in vivo* is

probably much lower due to slow diffusion of substances from the cross-linked matrix of the BDRA, as well as dilution and buffering of circulating body fluids. Additionally, the potential toxicity of residual monomers was further evaluated by histological analysis. Preclinical validation in abdominal subcutaneous (Figure. R7c) and dorsal skin wound (Figure. R7d) demonstrated that BDRAs, along with their corresponding residual precursors, caused only a milder inflammatory response than octyl CA at an early stage that gradually dissipated. Furthermore, there were no noticeable indications of damage or pathological changes of major organs (heart, liver, spleen, lung, and kidney) (Figure. R7e), confirming a minor inflammatory response and tolerable toxicity.

The relevant details have been supplemented in the revised manuscript (page 8, line 216-230) and as follows: “Furthermore, the potential toxicity of residual monomers of BDRAs was evaluated (Supplementary Fig. 12). The concentrations of residual precursors including MDO, HEMA, NHS, BPO and DMPT presented no potential cytotoxicity according to International Standard ISO 10993-5. Although the highest concentrations of residual HEA were found cytotoxic in the cell tests, preclinical validation in rat abdominal and dorsal implant models demonstrated that the BDRA adhesives, along with their corresponding residual precursors, caused only a milder inflammatory response than octyl CA at an early stage, and the inflammation gradually disappeared (Fig. 3g). As expected, the BDRAs was verified a less proliferating lymphocyte (CD3) and macrophage (CD68) than CA after 7 days (Fig. 3h). And the signs of inflammation for BDRA almost disappeared at 2 weeks, while obviously visible for CA (Fig. 3i). The BDRA adhesives did not cause any noticeable indications of damage or pathological changes of major organs, including the heart, liver, spleen, lung, and kidney, indicating tolerable toxicity and biocompatibility (Supplementary Fig. 13).”

Thank you again!

Reviewer #3:

Comments:

Current study has put forward the design of a programmed backbone-degradable robust adhesives via radical ring opening polymerization. The adhesive was claimed to exhibit superior adhesive strength toward diverse materials and tissues as well as tunable setting time and degradation rate. Tissue adhesive has been attracting attention due to its huge potential in clinical medicine. The adhesive system from Yang et al. has been highlighted on its capacity of promoting the penetration of adhesive molecule into substrate to achieve higher adhesion strength and allowing programmed degradability. These points could provide valuable clues for researches in tissue

adhesive materials. However, based on current results, the performance of designed adhesive was not comprehensively characterized and could hardly satisfy the high demand of Nature Communication on the aspect of innovation. Overall, I suggest a rejection for this manuscript. Associated problems have been listed as follow, hopefully to help you improve your manuscript.

Response:

We greatly appreciate the positive comments and helpful suggestions to improve the manuscript. We regret not clearly demonstrating the uniqueness and innovation of our work in the original manuscript.

Biodegradable strong adhesives with *in-situ* rapid adhesion are significantly important in technical and clinical applications. One of the primary challenges is achieving the strong adhesion *in-situ* under physiological environments (wet, room temperature, oxygen/air), and the adhesion strength (~kPa) of most reported adhesives is still unsatisfied for many fields, such as bone adhesion (which requires ~MPa of adhesion strength) [see Nature, 2019, 575, 169]. An additional challenge is to achieve biodegradability without compromising adhesion strength. Besides strong adhesion to tissue, an ideal tissue adhesive must be biodegradable to achieving initial mechanical integration and subsequent bio-integration, avoiding the secondary surgical removal. However, most strong adhesives are nondegradable. Nature-derived adhesives are generally biodegradable, but their adhesion strengths are one order of magnitude weaker than synthetic adhesives due to their intrinsic low cohesive strength [see Science, 2017, 358, 872-873]. Therefore, it is a contradictory problem for tissue adhesives to achieve high adhesion strength and biodegradability simultaneously.

Our work firstly proposed a universal strategy to prepare a backbone-degradable robust adhesive (BDRA) by *in-situ* radical ring-opening polymerization (rROP) to solve this problem effectively. rROP is a ring-opening polymerization initiated by radical as active species. With the tolerance to oxygen and moisture and free of metal-based catalysis, rROP can occur *in-situ* in the physiological environments. As the precursors of rROP, the hydrophobic cyclic ketene acetal (CKA) monomers and hydrophilic acrylate comonomers can form the polymer backbone with degradable ester groups through rROP, which endows it with good degradability. Additionally, the amphiphilic properties make the precursors with better wetting and penetration for the adherent tissues and subsequently form a covalently interpenetrating network with tissues through *in situ* rROP, which allows to achieve the high-strength adhesion.

The BDRAs exhibited strong adhesion to diverse materials even for wet biological tissue (e.g., wet bone >16 MPa, superior to the commercial cyanoacrylate (CA) super-adhesive ~4 MPa), and outperformed sixteen well-recognized adhesives including non-degradable category. The BDRAs avoid the drawbacks of cyanoacrylate super-adhesives in wet adhesion, and achieve good *in vivo* biodegradation, which different

from traditional acrylate adhesives and biodegradable polyester adhesives. This work will pioneer the *in-situ* rROP strategy for preparing adhesives, and transformative advances in the tunable synthesis of degradable robust adhesives will probably be comparable to the invention of CA super-adhesive.

Besides, we provided more comprehensive characterizations such as the thermal effects of the adhesives, cytotoxicity studies of the adhesive precursors, histological and staining analysis. Additionally, we have improved the *in vivo* study including the biodegradation of CA, observation of adhesive-tissue interaction at different time points after implantation, and an in-depth investigation of liver injury model and skull fracture model corresponding to soft tissue and hard tissue, respectively.

After the above revision, we believe that the importance, innovation, systematicity, and research depth of this article are sufficient for publication in the journal *Nature Communications*.

Question 1: Ring-opening polymerization has been employed in construction of tissue adhesives. The innovation of current one should be discussed compared with previous ones.

Response:

Thanks for the insightful suggestion. Radical ring-opening polymerization (rROP) of cyclic ketene acetals (CKAs) and acrylate comonomers combines the advantages of both ring-opening polymerization (ROP) and radical polymerization thereby allowing the robust production of polyesters coupled with the mild polymerization conditions of a radical-initiated process. Compared with the tissue adhesive by employing ROP of lactones, cyclic carbonates, or N-carboxyanhydrides initiated by anions or cations, the design of BDRA based on rROP has significant advantages:

(1) Free of metal-based catalysis, tolerance to oxygen and moisture, which allows for the *in-situ* polymerization and is crucial for applications in biological systems

(2) *In-situ rapid* formation of high-modulus and backbone-degradable polymer networks activated by the radical initiators like redox initiation system, avoiding the high temperature and long polymerization time of ROP initiated by anions or cations, thus achieving both degradability and high adhesion strength.

(3) Retaining the versatility and robustness of radical polymerization, ease of synthesis and functionalization, having diverse structures and performance with a wide adjustable window, such as mechanical performances, setting time and degradability.

We have reorganized in the section of Introduction (page 2, line 52-68), and the relevant details as follows:

“Achieving high adhesive strength and degradability requires combining two aspects:
(I) Well-permeated precursors cured to form a cross-linked network with good

mechanical properties to obtain robust cohesion and interfacial bonding. (II) The backbone of the in situ formed polymeric network has biodegradable units.

.....

Inspired by the synergistic roles of the hydrophilic and hydrophobic chains of mussel foot proteins, an in-situ radical ring-opening polymerization (rROP) of hydrophobic cyclic ketene acetal (CKA) monomers and hydrophilic acrylate comonomers in the physiological environment was first proposed to tunable synthesize a backbone-degradable robust adhesive (BDRA) (Fig. 1c). The rROP efficiently combines the simplicity and robustness of the radical polymerization of vinyl monomers with the controlled degradable chain of ring-opening polymerization of lactones, cyclic carbonates, or N-carboxyanhydrides³³, so it has been utilized to pre-synthesize biodegradable polymers for further biomedical applications, e.g., bio-degradable micelles/nanoparticles and functional polyester sealing materials^{34, 35}. Herein, the rROP is initiated by a redox system to in-situ form backbone-degradable polymer adhesive without being affected by environmental factors, including water (Fig. 1d). Noteworthy, the amphipathic BDRA precursors are well wetted before curing and achieve unexpectedly high diffusion-dominated interpenetration towards adherends with different surface energies, even for dense biological tissues. Ultrastrong adhesion is achieved upon the backbone-degradable covalent interpenetrating network solidified in a wide setting window ranging from seconds to hours (Fig. 1e). Benefiting from the flexibility and controllability of the rROP, the degradability and mechanical properties of the BDRAs can be customized on-demand, and their overall performance is superior to that of sixteen well-recognized adhesives³⁶⁻³⁹ (Fig. 1f and Supplementary Table 1), providing a broad spectrum of possibilities for biomedical engineering and medical applications in a facile and environmentally friendly manner.”

Question 2: The specific usage of designed adhesive was ambiguous in this study. No specific target tissue was mentioned. Despite that bone, skin, artery, liver and heart tissues were employed for assessment of adhesiveness, corresponding characterization methods remained on relatively superficial level.

Response:

We agree with these comments. According to these comments, on the basis of retaining the research content of bone, skin, artery, liver and heart tissues for showing the universality of BDRA for various hard and soft tissues, we have chosen a liver injury model and a skull fracture model for in-depth validation as specific applications of soft and hard tissues, respectively.

Liver injury model. BDRAs can effectively seal bleeding by providing fast and strong bio-adhesion. We have provided hemostasis time (~13 s) and gross observation

of wound closure and hemostasis for comparison with clinical treatment and commercial haemostatic products (see new Figs. 5a and 5c, Supplementary Fig. 25). Additionally, BDRA effectively sealed bleeding for liver injury and showed more stable adhesion compared to CA and the commercial fibrin glue Fibringluraas due to its strong bio-adhesion and adaptive modulus (figure R8a and Supplementary Video 5). Also, the blood loss (<60 mg) of hepatic perforation sealed by BDRA was nearly half that of the commercial haemostatic material Surgicel and comparable to that of Fibringluraas (Figures R8b-c). Besides, we supplemented tissue staining of the damaged rat liver after sealing by BDRA (MDO₁-HEA₁-NHS_{1/2}) and CA (Vetbond) for 1,7,14 days (Figures R8d).

Figure R8. Hemostasis and healing of liver injury. (a) Representative images for hemostasis of liver amputation injury treated with the BDRA (MDO₁-HEA₁-NHS_{1/2}) and CA glue. (b) Illustration of liver perforation model of rat. (c) Weight of blood loss from 4-mm diameter perforated wounds in a liver injury model. Data are presented as the means ± SDs, n = 3-5. (d) Representative H&E images of the damaged rat liver after sealing by BDRA (MDO₁-HEA₁-NHS_{1/2}) or CA (Vetbond) for 1, 7, 14 days, scale bar = 500 μm.

Rat skull fracture model. As shown in the Figure R9a, BDRAs (MDO-HEMA) accurately repositioned the skull fragments of rats within 3 minutes in the

physiological environment. There was significant bone tissue bridging in the BDRA group after 8 weeks, while bone nonunion was persisted in the blank group and the nondegradable CA group (Figure R9b). The degradable BDRA in fractures diminished during bone regeneration and ingrowth and almost disappeared over 8 weeks, while the CA was still significantly observed (Figure R9c). Quantitative analysis of regenerated bone regeneration revealed that the bone volume fraction (BV/TV) of biodegradable BDRA was the highest among all groups (Figure R9d). Consistent with the observation and quantitative analysis of the micro-CT results, biodegradable BDRA have more newly-formed bone islands and dense fibrous tissue (Figures R9e and R9f).

The relevant details were supplemented in the revised manuscript (page 11-12), referring to new Figs. 5f-j, and new Supplementary Fig. 26.

Figure R9. *In vivo* bone regeneration of rat skull fracture model. (a) Rapid fixation of the skull block with BDRA (MDO₁-HEMA₁). Scale bar, 5 mm. (b) Representative 2D Micro-CT images of rat skulls after fracture fixed by BDRA and Vetbond, and those without glue as a blank control. Scale bar, 1 mm. (c) Representative 3D CT reconstruction images of skull fracture. (d) Quantitative analysis of regenerated bone tissues by bone volume/total volume (BV/TV). (e) Representative H&E staining for regenerated bone tissues. The red stars represent glues, and blue triangles indicate the newly formed bone. Scale bar, 1 mm. (f) Representative Masson's trichrome staining for regenerated bone tissues. Scale bar, 1 mm.

Question 3: The degradation performance of cyanoacrylate adhesive should also be studied for comparison with BDRAs.

Response:

Thanks for the valuable advice. We have supplemented the degradation performance of the cyanoacrylate adhesive as shown in Figure R10. In detail, we soaked the cyanoacrylate adhesive (CA) in PBS at 37°C for *in vitro* hydrolysis. And we evaluated the *in vivo* biodegradation by subcutaneously implanting CA into rats. The degradation rates of CA were 8% after 16 weeks in PBS and 9% after 8 weeks of implantation, which were much lower than those of BDRA (~43% after 16 weeks in PBS and ~36% after 8 weeks of implantation).

Figure R10. (a), *In vitro* degradation rates of CA, PCL, PHEMA, and BDRAs (MDO₁-HEMA₁) based on the remaining weight percentage in PBS at 37 °C. (b), *In vivo* biodegradation behaviour of these adhesives implanted subcutaneously in rats.

The relevant details were supplemented as follows:

“The degradation rates of the composition-optimized BDRAs (MDO₁-HEMA₁) were 43% after 16 weeks in PBS and 36% after 8 weeks of implantation, which were superior to those of CA (8% and 9%), PCL (1% and 2%) and PHEMA homopolymer (18% and 3%).” --- In revised manuscript, page 8, line 202-203.

“PHEMA and CA (Vetbond) were prepared and tested in the same way and along with PCL were used as control groups.” --- In revised manuscript, page 16, line 508-509.

Question 4: *In vivo* investigation in this study could hardly fully characterize the performance of designed adhesive. The degradation of adhesive *in vivo* was accompanied by host tissue ingrowth and penetration. *In vivo* characterization of a degradable adhesive materials should include the observation of adhesive-tissue interaction at different time points after implantation.

Response:

Thanks for your thoughtful consideration and helpful suggestions. To depict the adhesive-tissue interaction of designed BDRA, the healing of damaged liver and skull

fracture was observed at different time points after implantation.

As shown in the Figure R8, biodegradable BDRA can be retained after hemostasis and has shown a weak hindrance to the healing of liver tissue damage compared to non-biodegradable CA adhesives after 14 days treatment. Besides, rat skull fracture model was used to illustrate the in vivo interaction between BDRA and hard tissue. After 2, 4, 8 weeks, micro-CT scanning was performed on bone tissue to depict the new bone formation. As shown in Figure R9b, there was significant bone tissue bridging in the BDRA group after 8 weeks, while bone nonunion was persisted in the blank group and the nondegradable CA group. The degradable BDRA in fractures diminished during bone regeneration and ingrowth and almost disappeared over 8 weeks, while the CA was still significantly observed (Figure R9c). Quantitative analysis of regenerated bone regeneration revealed that the bone volume fraction (BV/TV) of biodegradable BDRA was the highest among all groups at the same time points, with values of 31.4 %, 50.6%, and 59.6% at 2, 4, and 8 weeks, respectively. In contrast, the BV/TV of nondegradable CA (33.1%) was even lower than the blank group (~39.7%) after 8 weeks post-surgery, representing an obstacle to tissue healing (Figure R9d). Consistent with the observation and quantitative analysis of the micro-CT results, biodegradable BDRA could provide a spatial environment for fibrous and osseous tissue ingrowth, allowing for more newly-formed bone islands and dense fibrous tissue (Figures R9e and R9f). These results demonstrated the degradability of adhesives is essential for tissue healing.

As seen above, our work provides material insight into biodegradable robust tissue adhesives (BDRAs), and the benefits of degradability and strong adhesion to tissue healing was demonstrated. Future works are expected to further improve the tissue specificity of BDRAs and prove their efficacy for enhancing tissue repair and regeneration.

Thank you again!

Sincerely Yours,

Prof. Shifang Luan

State Key Laboratory of Polymer Physics and Chemistry,

Changchun Institute of Applied Chemistry,
Chinese Academy of Sciences,
Renming Street, Changchun 130022, P. R. China
Tel.: +86 431 85262159; fax: +86 431 85262109
E-mail: sfluan@ciac.ac.cn
Homepage: <http://luanshifang.polymer.cn>

Reviewers' Comments:

Reviewer #1:

Remarks to the Author:

Previously raised issues have been addressed very well by the authors. The reviewer has no additional concern.

Reviewer #2:

Remarks to the Author:

The authors have performed extensive revision of their manuscript and have addressed my comments (and those from the other reviewer) very seriously.

Overall, I can now recommend publication in Nature Commun

Reviewer #3:

Remarks to the Author:

Luan et al. has revised the previous manuscript and supplemented some crucial data for better demonstration of their design.

In specific, the introduction section has been revised to highlight the uniqueness and innovation of in-situ radical ring-opening polymerization (rROP) strategy of the adhesive. The rROP strategy was claimed to take the respective advantages of radical polarization and ring-opening polymerization. Corresponding data such as polymerization temperature and degradation rate was also supplemented to back up the statement.

In addition, in vivo investigation on adhesiveness and degradability was also added. However, there is still one question to solve. Is the adhesive designed for bonding load-bearing or non-load-bearing bones? If it's for fixing load-bearing bones, the mechanical performance of adhesion during degradation process should also be investigated since degradation and adhesive strength could be a contradictory couple under in vivo condition.

Point-to-point response to reviewers

Reviewer #1:

Comments: Previously raised issues have been addressed very well by the authors. The reviewer has no additional concern.

Response: Thanks for the reviewer's positive comment.

Reviewer #2:

Comments: The authors have performed extensive revision of their manuscript and have addressed my comments (and those from the other reviewer) very seriously. Overall, I can now recommend publication in Nature Commun.

Response: Thanks for the reviewer's positive comment.

Reviewer #3:

Comments: Luan et al. has revised the previous manuscript and supplemented some crucial data for better demonstration of their design. In specific, the introduction section has been revised to highlight the uniqueness and innovation of in-situ radical ring-opening polymerization (rROP) strategy of the adhesive. The rROP strategy was claimed to take the respective advantages of radical polarization and ring-opening polymerization. Corresponding data such as polymerization temperature and degradation rate was also supplemented to back up the statement. In addition, *in vivo* investigation on adhesiveness and degradability was also added. However, there is still one question to solve. Is the adhesive designed for bonding load-bearing or non-load-bearing bones? If it's for fixing load-bearing bones, the mechanical performance of adhesion during degradation process should also be investigated since degradation and adhesive strength could be a contradictory couple under *in vivo* condition.

Response:

We genuinely appreciate the positive comments on our manuscript. And the suggestions raised by the reviewer are important and valuable.

BDRAs are designed as a universal strategy applicable for both soft and hard tissues adhesion. They can theoretically be customized to hard tissue, including both load-bearing and non-load-bearing bone. In this work, a skull full-thickness defect model is used as a representation of non-load-bearing bone for *in vivo* demonstration in the adhesion of hard tissues. Our concept is validated by the results that the adhesive can stably fix the bone fragments (Figure R1a-b), while the bone tissue of the defect gradually grows in with the degradation of the adhesive (Figure R1c-f).

Figure R1. *In vivo* bone regeneration of rat skull fracture model. (a) Rapid fixation of the skull block with BDRA (MDO₁-HEMA₁). Scale bar, 5 mm. (b) Representative 2D Micro-CT images of rat skulls after fracture fixed by BDRA and Vetbond, and those without glue as a blank control. Scale bar, 1 mm. (c) Representative 3D CT reconstruction images of skull fracture. (d) Quantitative analysis of regenerated bone tissues by bone volume/total volume (BV/IT). (e) Representative H&E staining for regenerated bone tissues. The red stars represent glues, and blue triangles indicate the newly formed bone. Scale bar, 1 mm. (f) Representative Masson's trichrome staining for regenerated bone tissues. Scale bar, 1 mm.

As our work progressed, we realize that the BDRA strategy might make particular sense for load-bearing bone applications due to its strong adhesion strength and degradable properties [see *Adv. Mater.*, 2020, 32, e1907491.]. Although the degradation and mechanical properties of the adhesive itself are contradictory, the degradable adhesive provides an opportunity for the growth and reconstruction of biological tissues. And the ingrown tissue, especially for high-modulus bone tissue, can provide biomechanics to compensate for the mechanical attenuation caused by the degradation of the adhesive [see *Nat. Rev. Mater.*, 2020, 5, 584.]. However, this process needs to be customized for specific load-bearing bone applications and validated by long-term *in vivo* observations.

Indeed, biodegradable robust bone adhesives represent a highly sought-after alternative for bone fracture fixation, especially for complicated bone fractures where conventional methods including metal fixation are ineffective [see *Nat. Rev. Dis.*

Primers., 2021, 7, 57; *Mater. Today. Bio.*, 2023, 19, 100599.]. Based on the in-depth research and exploitation of this *in-situ* rROP strategy, we will further optimize the performance of BDRAs in the specific applications of load-bearing bone. The visionary suggestion provided by the reviewer regarding the mechanical performance of adhesion during the degradation process will be investigated in the future work as soon as possible.

Best regard!

Sincerely Yours,

Prof. Shifang Luan

State Key Laboratory of Polymer Physics and Chemistry,

Changchun Institute of Applied Chemistry,

Chinese Academy of Sciences,

Renming Street, Changchun 130022, P. R. China

Tel.: +86 431 85262159; fax: +86 431 85262109

E-mail: sfluan@ciac.ac.cn

Homepage: <http://luanshifang.polymer.cn>

Reviewers' Comments:

Reviewer #3:

Remarks to the Author:

The authors have supplemented necessary data and provided proper answers to my questions. I suggest current version of manuscript could be accepted for publication.

Point-to-point response to reviewers

Reviewer #3:

Comments: The authors have supplemented necessary data and provided proper answers to my questions. I suggest current version of manuscript could be accepted for publication.

Response: Thanks for the reviewer's positive comment.

Best regard!

Sincerely Yours,